# Genome-scale community modelling reveals conserved metabolic cross-feedings in epipelagic bacterioplankton communities

Nils Giordano [1,4], Marinna Gaudin [1,4], Camille Trottier [1], Erwan Delage[1], Charlotte Nef[2], Chris Bowler [2,3] & Samuel Chaffron [1,3] ✉

Marine microorganisms form complex communities of interacting organisms that influence central ecosystem functions in the ocean such as primary production and nutrient cycling. Identifying the mechanisms controlling their assembly and activities is a major challenge in microbial ecology. Here, we integrated *Tara* Oceans meta-omics data to predict genome-scale community interactions within prokaryotic assemblages in the euphotic ocean. A global genome-resolved co-activity network revealed a significant number of inter-lineage associations across diverse phylogenetic distances. Identified co-active communities include species displaying smaller genomes but encoding a higher potential for quorum sensing, biofilm formation, and secondary metabolism. Community metabolic modelling reveals a higher potential for interaction within co-active communities and points towards conserved metabolic cross-feedings, in particular of specific amino acids and group B vitamins. Our integrated ecological and metabolic modelling approach suggests that genome streamlining and metabolic auxotrophies may act as joint mechanisms shaping bacterioplankton community assembly in the global ocean surface.

Marine microbes constantly interact among each other and with their environment, forming complex and dynamic networks. These communities and their interactions play crucial ecological and biogeochemical roles on our planet, forming the basis of the marine food web, sustaining biogeochemical cycles in the ocean, and regulating climate[1]. Complex networks of trophic interactions, mediated through metabolic cross-feeding and ecological successions, can influence the nature of microbial interactions (e.g., mutualism or competition), in space and time, and thus significantly shape microbial community assembly[2]. Expanding our understanding of microbial trophic interactions is fundamental given their capacity to modulate ecological niches[3], constrain microbial biogeography[4], drive microbial diversification[5],

and modulate the eco-evolutionary dynamics of microbial communities[6]. Because most microbes are difficult to isolate and cultivate in lab-controlled environments[7], and given the large diversity of molecules that can be excreted into the environment (e.g., waste metabolites, secondary metabolites, exoenzymes, siderophores), we are just starting to grasp the complexity and diversity of microbial interactions and cross-feeding relationships existing in nature[8]. In particular, we lack a mechanistic understanding of metabolic auxotrophy and its role in constraining marine microbial community composition and assembly[9].

While species co-occurrence networks are useful tools to model the large-scale structure of microbial communities[10] and to resolve biome-specific ecological associations[11], these approaches are

[1]Nantes Université, École Centrale Nantes, CNRS, LS2N, UMR 6004, F-44000 Nantes, France. [2]Institut de Biologie de l'École Normale Supérieure (IBENS), École Normale Supérieure, CNRS, INSERM, PSL Université Paris, F-75016 Paris, France. [3]Research Federation for the Study of Global Ocean Systems Ecology and Evolution, FR2022/Tara Oceans GOSEE, F-75016 Paris, France. [4]These authors contributed equally: Nils Giordano, Marinna Gaudin. ✉e-mail: samuel.chaffron@cnrs.fr

inherently limited since correlation metrics do not provide evidence for direct biotic interactions, and do not allow to disentangle true biotic interactions from environmental preferences (niche overlap)[12]. Thus, we still lack a comprehensive and mechanistic understanding of biotic and abiotic interactions shaping community assembly of microbial communities. Ecosystem modelling approaches are therefore needed to capture and predict emergent properties resulting from complex interactions within microbial communities, such as resilience, niche space, and biogeography, that shape microbial communities and ecosystems[13]. Recent experimental work has demonstrated the significant impact of underlying cross-feeding metabolic networks in shaping community assembly[14] and ecological successions[15] in synthetic microbial communities. Using microbial community assembly experiments in soil, coupled with a simple resource-partitioning model, functional convergence was shown to be mainly driven by emergent metabolic self-organization, while taxonomic divergence seemed to arise from multi-stability in population dynamics[14]. In another system, coculture experiments of a marine microbial community able to degrade chitin demonstrated the hierarchical preferences for specific substrates, underlining the sequential colonization of metabolically distinct groups, and identifying hierarchical cross-feedings shaping the dynamics of community assembly[15].

Recent large-scale environmental surveys of marine microbial ecosystems (e.g., *Tara* Oceans[16], Malaspina[17], Bio-GO-SHIP[18], BioGEOTRACES[19]) have generated large volumes of metagenomics data that enable the reconstruction of genomes from uncultivated species referred to as Metagenome-Assembled Genomes (MAGs)[20,21]. Together with whole genome sequences (WGS) from cultured organisms and single amplified genomes (SAGs) from single cell isolates, these resources have been used to expand our knowledge of microbial diversity in the ocean, but have also demonstrated that a large fraction of the diversity remains to be explored[22,23]. In this context, genome-resolved metagenomics provides an opportunity to enrich co-occurrence signals with genetic information from genomes and functional information from genome-scale metabolic models. Integrating this knowledge into association networks can inform us about the functional self-organisation of microbial communities[24], contribute to our understanding of species interactions mechanics, and identify general ecological laws that structure microbial communities. While community metabolic modelling approaches have recently been applied to study the self-organisation of microbial ecosystems[25] and to gain insights into molecular mechanisms of interactions in soil[26], wastewater[27], and gut microbiome communities[28], few studies so far have focused on the modelling of marine plankton ecosystems[15,29], and were limited to specific single communities.

Here, we describe an integrated ecological and metabolic modelling approach (Supplementary Fig. 1) with the goal to delineate metabolically cohesive consortia underlying genes-to-community assembly and ecosystem functioning at global scale[30]. We combined co-activity ecological information inferred from meta-omics with community metabolic simulations using genome-scale metabolic models to uncover putative biotic interactions mediated by metabolic cross-feedings among marine prokaryotic genomes. Through a multi-omic approach integrating *Tara* Oceans metagenomic and metatranscriptomic datasets, we inferred a global ocean genome-resolved ecological network from whole-genome transcriptomic activities. We used general genomic scaling laws[31] as a framework to characterise the functional content of co-active environmental genomes, and identified functional gene categories likely driving metabolic dependencies. We then reconstructed genome-scale metabolic models and uncovered putative cross-feeding interactions within co-active consortia through the use of community-level metabolic modelling.

## Results and discussion
### Genomic scaling laws reveal features of uncultivated marine prokaryotic genomes

To build a comprehensive catalogue of marine prokaryotic genomes, we collected and assembled public whole-genome sequences (WGS) from marine prokaryote isolates[32], single-amplified genomes[22] (SAGs), as well as previously reconstructed MAGs from hundreds of public metagenomes including *Tara* Oceans metagenomes[21]. This integrated marine prokaryotic genome database counted 7658 non-redundant species-level representative genomes (delineated by a 95% ANI threshold over 60% of genome length, see methods and Supplementary Data 1). Herein, we only considered genomes meeting sufficient quality standards ($n = 5678$), that is High-Quality (HQ) MAGs (≥90% complete and ≤5% contamination), Medium-High-Quality (MHQ) MAGs (≥75% complete and ≤10% contamination), and Medium-Quality (MQ) MAGs (≥50% complete and ≤25% contamination). HQ and MHQ MAGs were not significantly different from WGS genomes in terms of gene density (Supplementary Data 2). A phylogeny of these genomes was established using domain-specific marker genes of the Genome Taxonomy Database (GTDB)[33], highlighting a total of 107 phyla (with unclassified) including highly represented phyla in marine environments, such as Proteobacteria, Bacteroidetes, Actinobacteria, and Cyanobacteria[34] (Fig. 1a).

In biology, scaling relationships and scaling laws are numerous (e.g., Kleiber's law of metabolic rate scaling with body mass in birds and mammals) and have been studied extensively[35]. For bacteria, within prokaryotic genomes, the number of genes in most high-level functional categories (regrouping related gene functions into broad functional categories such as COG and KEGG BRITE functional hierarchies) has been shown to scale as a power-law to the total number of genes in the genome[36]. A potential explanation for these observed scaling laws among microbial genomes is a conserved average duplication rates within each functional category. In addition, these genomic scaling laws have been shown to be conserved across microbial clades and lifestyles, supporting the observation that they are universally shared by all prokaryotes[31]. However, these genomic scaling laws have never been investigated within uncultured genomes so far. Here, we thus revisited this universal law for environmental marine genomes (MAGs and SAGs). To ensure a sound and fair comparison between WGS and environmental genomes, we limited our analysis to MHQ genomes, which displayed a similar gene density as compared to WGS (Supplementary Fig. 2b). We showed that MHQ genomes did actually fit the same law as WGS genomes (Fig. 1b), and thus limited all subsequent analyses to MHQ genomes only. This analysis also revealed that MHQ MAGs were systematically smaller in genome size and number of predicted CDS as compared with WGS genomes. This observation is coherent with the assumption that a large fraction of naturally occurring marine genomes have likely adapted to oligotrophic surface ocean specific lifestyles through genome streamlining[37]. Scaling laws are a powerful and sound way to compare functional potentials among genomes as they allow to reveal unexpected deviations from the general trend while taking into account genome size variations. Investigating the genomic scaling laws for high-level functional categories (see methods), we showed that this adaptation has differentially impacted a majority of metabolic functions (75%) within environmental genomes (MAGs and SAGs), but with notable increase potential in MAGs for metabolic functions likely playing a key role in mediating biotic interactions, such as for xenobiotic degradation, terpenoid and polyketide metabolism, as well as lipid metabolism, but a decrease potential to synthesize cofactors and vitamins (Supplementary Fig. 3 and Supplementary Data 3). This decreased metabolic potential for cofactors and vitamins in environmental genomes likely reflects the importance of syntrophic metabolism, such as metabolism of essential enzyme cofactors[38], and associated bacterial traits for microbial interactions[39], to sustain microbial life in the surface ocean that is largely depleted in B vitamins[40].

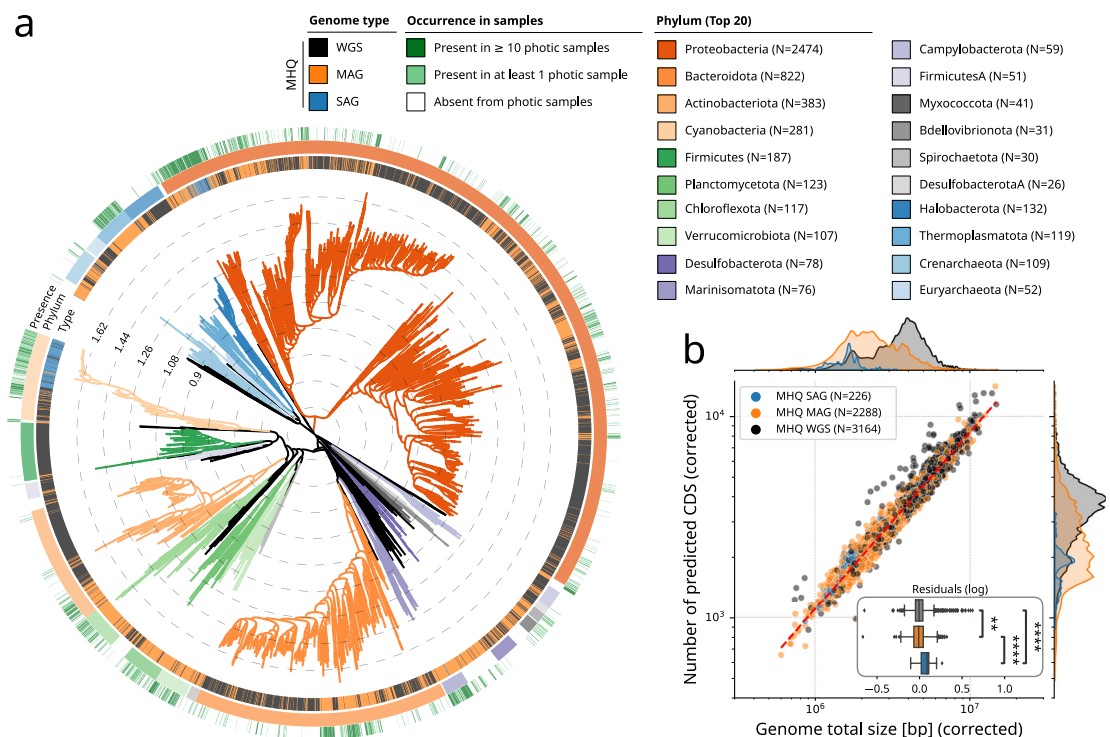

**Fig. 1 | A database of marine bacterial and archaeal genomes from isolates and uncultivated genomes reconstructed from marine metagenomes.**
**a** Phylogenetic tree of the database of marine genomes ($N = 7658$) dereplicated at species level (95% Average Nucleotide Identity or ANI). Reference genomes (WGS) were obtained from MarRef, MarDB, and aquatic progenomes, while Metagenome-Assembled Genomes (MAGs) and Single-Amplified Genomes (SAGs) were also obtained from different studies (see "Methods"). A total of 107 phyla (including unclassified) were detected (the top 20 most represented phyla are highlighted).
**b** A comparison of genome size and number of predicted CDS, both corrected by genome completeness (division by completeness), revealed that a genome scaling law is conserved for High and Medium-High Quality (HQ and MHQ) genomes (completeness ≥75% and contamination ≤5%), and that MAGs overall displayed significantly smaller genomes ($p = 5.58 \times 10^{-194}$, two-sided Mann–Whitney U test on log-transformed distributions). The box extends from the lower to upper quartile values of the data (Q1 and Q3), with a line at the median (Q2). The whiskers extend from the box to show the range of the data and are defined as follows: where IQR is the interquartile range (Q3-Q1), the upper whisker will extend to last data point less than $Q3 + 1.5 \times IQR$. Similarly, the lower whisker will extend to the first data point greater than $Q1 – 1.5 \times IQR$. Beyond the whiskers, data are plotted as individual points.

## Abiotic factors shaping genome community composition and activity

Next, we mapped *Tara* Oceans metagenomics and metatranscriptomics sequencing reads from surface (SRF) and deep chlorophyll maximum (DCM) samples (N = 118) onto our genome collection to generate a global ocean abundance and expression profiling of microbial communities in relationship with abiotic environmental factors (see Supplementary Data 4). Average mapping rates were 16.0% and 12.3% for metagenomes and metatranscriptomes, respectively (Fig. 2a and Supplementary Fig. 4). The presence, abundance, and activity of a given genome was determined as follows: first, the occurrence of a genome was determined by its horizontal metagenomic coverage of minimum 30%; second, its abundance was computed using its vertical metagenomic coverage normalised by its genome length. Finally, its activity corresponded to the ratio of its vertical metatranscriptomic coverage over its vertical metagenomic coverage (see methods for details). Using the same *Tara* Oceans dataset, gene and transcript abundances have previously been shown to be highly correlated[41]. Here, we observed an overall relatively good concordance between genome-wide abundance and expression (Spearman rho=0.68, $p = 0$), albeit a number of genomes displayed lower genome-wide expression levels (Fig. 2b), highlighting the complementary information brought by genome expression signals computed here. Thus, this observation prompted us to compute genome-wide activities, integrating abundance and expression levels at the genome scale (see methods). Principal Coordinates Analyses (Fig. 2c) did not reveal a clear structuration of community genome

assemblages and activities by ocean basin, but allowed us to identify abiotic factors driving community composition in abundance and activity. Genome community composition was mainly driven by temperature, pH, and Photosynthetically Available Radiation (PAR), while genome community activity was mainly driven by temperature, phosphate ($PO_4$) and iron concentrations (see methods and Supplementary Data 5).

Temperature has previously been shown to be one of the main factors constraining epipelagic bacterioplankton community composition[34], which is confirmed here for both genome-wide community abundance and activity. The effect of (small) pH changes on marine microbial communities has mainly been shown experimentally[42,43], but often not considering the natural variability of pH in the surface ocean[44]. Other studies have reported minor effects of acidification on the productivity of natural picocyanobacteria assemblages[45]. Here, the observed association between genome community composition and pH could partly be explained by seasonal variability encountered during global sampling. While genome community activity was principally associated to temperature, distinct environmental factors, namely $PO_4$ and iron concentrations, were also significantly associated to community activity. This observation emphasises the major role of nutrients and/or cofactors (co-)limitations in structuring global ocean microbial activity[46,47].

## Biotic drivers of genome activity community structure
While abiotic factors are known to be significant drivers of microbial community structures in the ocean, biotic factors (such as

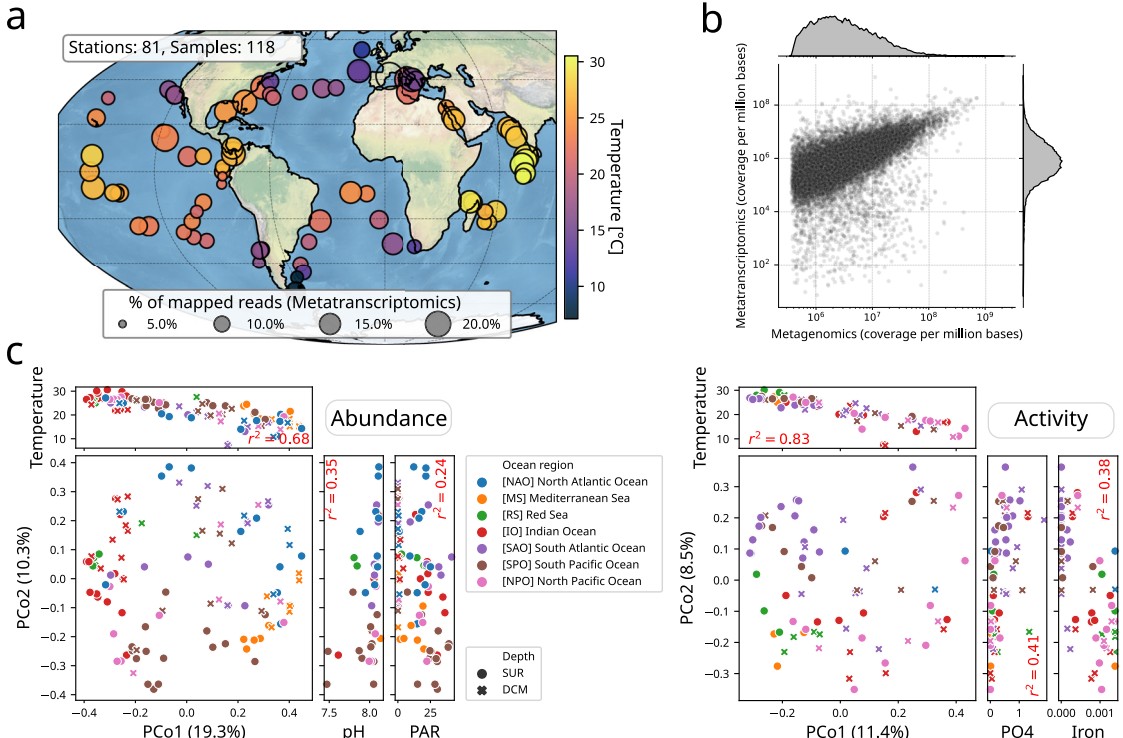

**Fig. 2 | Genome-wide abundance and activity profiling of marine prokaryotic genomes in the global surface ocean. a** World map of *Tara* Oceans sampling stations (N = 81) for which euphotic (SRF and DCM) metatranscriptomes are available for a prokaryote-enriched size fraction (0.22–3 µm). The percentage of mapped RNA reads are depicted for each euphotic sample (N = 118). **b** Genome-wide abundance and expression were significantly associated (Spearman rho=0.68, p = 0). albeit a number of genomes display lower expression levels. **c** Principal Coordinates Analyses (PCoA) for genome community abundances (across 107 samples with metagenomics data) and activities (across a subset of 71 samples with both metagenomics and metatranscriptomics data available). Genome community abundance and activity (PCo1) are significantly associated with temperature. Community abundance (PCo2) is also associated with pH and Photosynthetically Available Radiation (PAR), while community activity is associated with PO4 and iron concentrations.

competition, parasitism, or mutualism) are expected to play an equally important role[48], though the latter are more difficult to study in natural communities. Microbial association networks are useful abstractions that represent potential biotic interactions and capture emergent properties (e.g., connectivity, functional redundancy) that result from these putative interactions[49]. But so far, most studies have been limited to the organismal level by predicting these ecological associations using taxonomic marker genes (e.g., 16 S and 18 S rRNA genes). Integrating genomic information into association networks can be particularly useful to draw and test hypotheses about the functional self-organisation of microbial communities[24]. Here, we went beyond by inferring a global ocean association network from genome activities that were inferred by integrating genome-wide abundance and transcript levels (here activity refers to a genome-wide ratio between transcript and genomic vertical coverages, as described above, and see methods for details). Thus, we used samples with both metagenomics and metatranscriptomics data available to compute genome-wide co-activity. We make the general assumption that a co-activity signal is a better proxy to capture biotic interactions as compared to co-abundance, given the latter is an integration of all past metabolic activities that cannot identify microbial cells that were actually transcriptionally active at sampling time. In other words, we expect genome-wide co-activity (integrating abundance and transcript levels) to be more sensitive as it inherently has a better time-resolution when searching for microbial interactions.

We inferred a genome-resolved co-activity network using the dedicated probabilistic learning algorithm FlashWeave (FW, see methods) that can efficiently detect and remove undirect associations among features[50]. FW first infers a global correlation network resulting from significant partial correlation tests with Fisher's z-transformation ("sensitive" mode) between genome pairs co-activity profiles. Next, FW uses a local-to-global learning algorithm that implements conditional independence tests to detect and remove indirect associations. The resulting network integrates only direct associations, whose edges are weighted by the partial correlation strength. This genome-resolved co-activity network was significantly different than the corresponding genome-resolved co-abundance network, with a higher number of edges in co-activity, and only a small fraction of shared edges (3%) (Supplementary Fig. 5). This strong difference between both networks can reflect the distinct information carried out by abundance and activity profiles, but can also be partially explained by the heuristics-based inference of direct associations as implemented in FW. The co-activity network revealed a larger number of significant positive associations across diverse phylogenetic distances (PD), while negative associations were mainly observed between phylogenetically distant genomes (Fig. 3a). In addition, the distributions of phylogenetic distances for negative and positive associations were significantly different (Mann-Whitney U-test with Bonferroni correction, $p = 1.494 \times 10^{-30}$). It also revealed two distinct types of positive associations: relative phylogenetically close associations (0 < PD < 1) that likely reflected niche overlap, and phylogenetically distant associations (PD ≥ 1) likely reflecting a higher potential for cross-feeding interactions[51]. As previously reported for co-existing genomes across various biomes[24], co-active genomes tended to be functionally closer (both in terms of encoded KO genes and expressed KO genes, see methods) than expected at random (Mann–Whitney U-test with Bonferroni correction, $p = 1.187 \times 10^{-25}$). This observation may reflect the

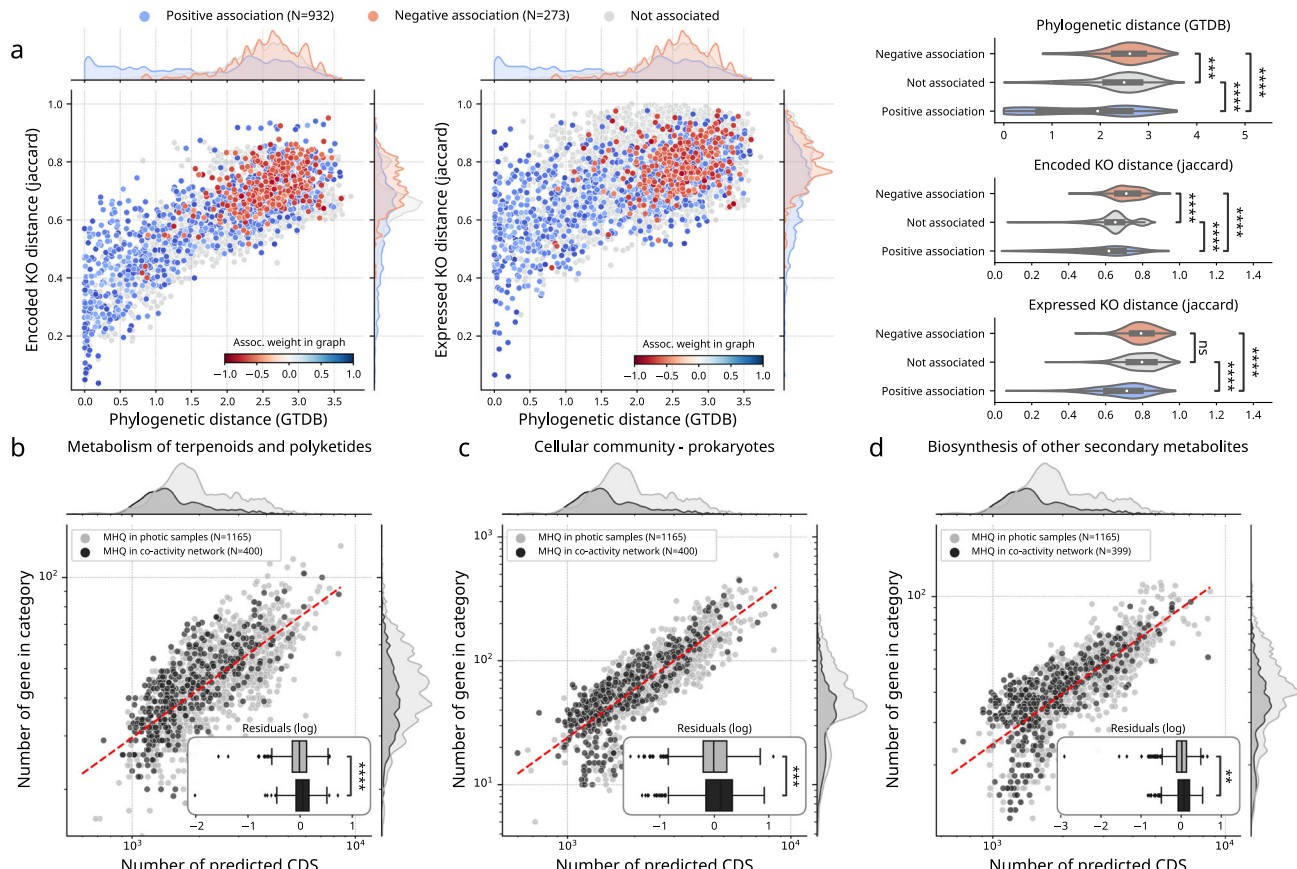

**Fig. 3 | A genome-resolved co-activity network reveals biotic factors shaping marine prokaryotic community structure. a** A genome-resolved co-activity network was inferred from genome-wide activities in euphotic samples, and revealed a larger number of significant positive associations between genomes across diverse phylogenetic distances. Based on encoded KO gene presence/absence, and expressed KO gene presence/absence, using Jaccard distances between genomes as a proxy for functional distance (using KEGG), co-active genomes were functionally closer than expected at random ($p = 1.187 \times 10^{-25}$, two-sided Mann–Whitney U test on log-transformed distributions). **b–d** Scaling laws in the functional content of genomes highlighted broad metabolic categories enriched in co-active genomes versus genomes detected as active in samples. Notably, co-active genomes displayed a higher functional potential for terpenoid and polyketide metabolism ($p = 1.46 \times 10^{-7}$, two-sided Mann–Whitney U test on log-

transformed distributions), for cellular community metabolism (quorum-sensing and biofilm formation, $p = 4.00 \times 10^{-4}$, two-sided Mann–Whitney U test on log-transformed distributions), and for the biosynthesis of other secondary metabolites ($p = 4.73 \times 10^{-9}$, two-sided Mann–Whitney U test on log-transformed distributions). Dashed-red lines are the best linear fit on a log-log scale (parameters given in Supplementary Data 3). The box extends from the lower to upper quartile values of the data (Q1 and Q3), with a line at the median (Q2). The whiskers extend from the box to show the range of the data and are defined as follows: where IQR is the interquartile range (Q3-Q1), the upper whisker will extend to last data point less than Q3 + 1.5 × IQR. Similarly, the lower whisker will extend to the first data point greater than Q1–1.5 × IQR. Beyond the whiskers, data are plotted as individual points. See Supplementary Data 6 for a complete list of functions enriched in co-active genomes.

impact of ecological preferences or niche overlap on evolution, that could be explained by adaptation to a same niche and/or by potential higher rates of horizontal gene transfer (HGT) in specific biomes[52]. Marine co-active genomes also tended to be smaller in size as compared to detected but non-co-active genomes, although displaying similar gene densities as assessed by genomic scaling laws (Supplementary Fig. 6), and despite the fact that most genomes detected in photic samples corresponded to MAGs (Fig. 1a) overall smaller in size (Fig. 1b). In addition, comparative genomics analyses based on scaling laws allowed us to take into account genome size (see methods). By identifying deviations from the power law (or linear law in log-log scale), we used scaling laws as a tool to properly identify enriched or depleted metabolic potentials within groups of genomes. In other words, to identify if a genome or group of genomes harboured fewer or more genes in a category than what would be expected given its size. A linear least-squares regression was performed on a log-log scale, and the distribution of residuals for each category (difference between actual y-axis value and expected y-axis value) were compared. When significantly different, these distributions indicated that the two

categories might not follow the same scaling laws. This revealed that co-active genomes displayed (in proportion) a higher metabolic potential for lipid, carbohydrate, and amino acid metabolism (Supplementary Fig. 7 and Supplementary Data 6), but also for terpenoids and polyketides, quorum-sensing and biofilm formation, as well as for secondary metabolite biosynthesis (Fig. 3b–d). Overall, these enriched genomic potentials in co-active genomes point towards key metabolic functions for energy harvest and storage (i.e., lipid, carbohydrate and amino-acids metabolism), likely key in nutrient-limited regions of the global ocean[47]. But they also underline key genomic enriched potential (i.e., antimicrobials and quorum-sensing) of marine genomes likely prone to a wide diversity of biotic interactions[39].

## Higher metabolic interaction potential in co-active bacterioplankton communities

To go beyond correlation-based and enrichment analyses and move towards a mechanistic understanding of marine microbial community functioning, we sought to model the community metabolism of co-active marine microbial genomes. To do this, we first

reconstructed genome-scale metabolic models for each MHQ genome (WGS or MAGs) using CarveMe[53] and quality checked them using MEMOTE[54] (Supplementary Materials). We then used Species Metabolic Coupling Analysis (SMETANA), a constraint-based technique commonly applied for modelling interspecies dependencies in microbial communities[55]. Here, SMETANA was used to compute several interaction scores (community-wide or pairwise) to predict metabolic interaction potential and reveal possible metabolic exchanges and cross-feedings within delineated communities of co-active genomes. Notably, the Metabolic Resource Overlap (MRO) score quantifies how many species in a given community compete for the same metabolites, and the Metabolic Interaction Potential (MIP) score quantifies how many metabolites can be shared between species to decrease their dependency on external resources. In addition, the SMETANA score, integrating several metrics to estimate the metabolic dependencies within a given community, was used to evaluate the probability of each potential cross-feeding interaction among identified co-active genome communities. Here, we analysed co-active genome communities identified by clustering the global co-activity network using the Markov clustering algorithm (see methods for details).

Overall, we observed a negative association between the MRO score and the mean community phylogenetic distance (Pearson $R^2 = 0.31$, $p = 4.16 \times 10^{-8}$, Supplementary Fig. 8), showing that, as expected, phylogenetically closer co-active genome communities tended to display a higher metabolic resource overlap, and thus a higher potential for competition. Co-active genome communities also displayed an overall lower MIP score as compared with random communities (Mann-Whitney U test, $p = 1.45 \times 10^{-17}$, Supplementary Fig. 9a). Nevertheless, both community-wide (MIP) and pairwise (SMETANA score) scores of metabolic interactions are significantly driven by the size of communities under consideration (Supplementary Fig. 9b), which we thus normalised by co-active community size, as previously done and reported[55]. Following this normalisation and despite overall higher MRO scores and mean community phylogenetic distance, co-active genome communities displayed a higher potential for metabolic interactions as compared with randomly assembled communities (Fig. 4a). These results show that metabolic cross-feeding interactions can occur across a large spectrum of phylogenetic and functional distances, suggesting that metabolic dissimilarity is one among other factors determining the establishment of cross-feeding interactions among bacteria[51].

Given the diverse phylogenetic distances observed among co-active genomes (Fig. 3a) and associated co-active communities (Supplementary Fig. 8), we sought to delineate distinct community types of co-active genomes in a non-supervised fashion (see "Methods"). Using this approach, we distinguished four types of communities of co-active genomes: randomly-assembled communities, largely composed of genome communities with a high mean phylogenetic distance (PD) and a low metabolic cross-feeding potential (CP) score (HPD and LCP), which we used as a reference to define three other community types corresponding to two communities with a Low-PD (LPD) and High- or Low-CP (H/LCP), and a third community with High-PD (HPD) and High-CP (HCP) (Fig. 4b). These four co-active genome community types displayed distinct taxonomic compositions, with LPD-HCP communities mainly composed of Gamma- and Alphaproteobacteria, while HPD-HCP were more diverse including genomes from classes Nitrososphaeria, Marinisomatia, Dehalococcoidia, Alphaproteobacteria, and Acidimicrobiia (Supplementary Fig. 10). To quantify how the functional potential of each community was shared between genomes, we used a proxy of the well-established Gini index (see methods). A Gini index of 0 can be interpreted as a perfect overlap between the functions of all members of the consortium, while a Gini index of 1 would be the extreme situation where a single member of the consortium displays all detected KO functions. As anticipated, both HPD

communities (orange and pink) were more dissimilar to respective LPD communities (blue and green) with regards to their encoded metabolism proxied by their functional Gini coefficient from KO genes occurrence profiles (Fig. 4c). Here, we hypothesised that these four community types displayed distinct signatures of metabolic exchanges and cross-feedings, which we analysed in details below.

## Key metabolic cross-feedings driving bacterioplankton community assembly

To further explore and identify molecular mechanisms driving these global patterns of predicted metabolic interactions, we analysed predicted metabolic exchanges within the four co-active genome community types delineated above. Both HPD-HCP and LPD-HCP communities were predicted to have a higher potential exchange in specific metabolites as revealed by a NMDS analysis of broad metabolic categories (see methods) preferentially exchanged within each community type (Fig. 5a). Here, the first two dimensions of co-variation (Dim1 and Dim2) highlighted amino acids (AAs), B-vitamins, organosulphur compounds, aliphatic amines, n-alkanals, and aromatics as metabolic categories most preferentially exchanged within HPD-HCP and LPD-HCP community types (Fig. 5b). Despite large differences in mean PD within these communities, preferentially exchanged metabolic categories appeared to be conserved in HPD-HCP and LPD-HCP community types, suggesting these predicted metabolic exchanges may be ancient and evolutionarily conserved[13], although we cannot exclude some of these metabolites might be metabolic wastes that could be exported to the environment as a stress response. Nevertheless, this observation raises a key question regarding which evolutionary mechanisms can actually stabilize metabolic cross-feedings within natural microbial communities[56]. Although little is known about the coevolutionary consequences of cooperative cross-feeding, stable coevolution is expected to increase productivity in cross-feeding communities, which has been corroborated by experimental evidence[57].

Investigating the biogeography of HCP communities (Supplementary Fig. 11) revealed that they were detected at global scale (i.e., across all samples) but displaying different degrees of regionalization (i.e., no single HCP community is detected across all samples). We observed a structuration along a latitudinal gradient of these communities, in relation to temperature, as previously reported using *Tara* Oceans amplicon sequencing data[11]. While HCP communities were associated to relatively different concentrations of Chlorophyll A and different levels of Net Primary Production (VGPM model), it should be noted that most ocean regions sampled during *Tara* Oceans corresponded to low productivity regions / oligotrophic zones. Zooming in broad metabolic categories, we identified specific metabolites predicted to be preferentially exchanged within all four community types (Fig. 5c). When considering inorganic compounds for community metabolic modelling, most preferentially exchanged compounds among all community types were phosphate and iron cations (Supplementary Fig. 12), likely due to the essential uptake of these limiting nutrients and co-factors in the ocean[46]. Thus, in order to focus on actual biotic metabolic exchanges predicted, we did not consider inorganic compounds as previously done in other studies[25].

Considering detailed predicted metabolic exchanges (using SMETANA sum scores) we identified compounds that were preferentially exchanged within each community type (Fig. 5c and Supplementary Data 7). In particular, acetaldehyde, benzoate, thiamine (vitamin B$_1$), ethanol, and L-glutamate were significantly more frequently exchanged within LPD-HCP communities, while in HPD-HCP communities exchanges of benzoate, thiamine, L-arginine, as well as D-glucose and D-ribose were significantly more prevalent (Supplementary Data 8). The relative importance of predicted AA exchanges, and in particular biosynthetically costly AAs (e.g., methionine, lysine, leucine, arginine), likely reflects the key role of syntrophic interactions

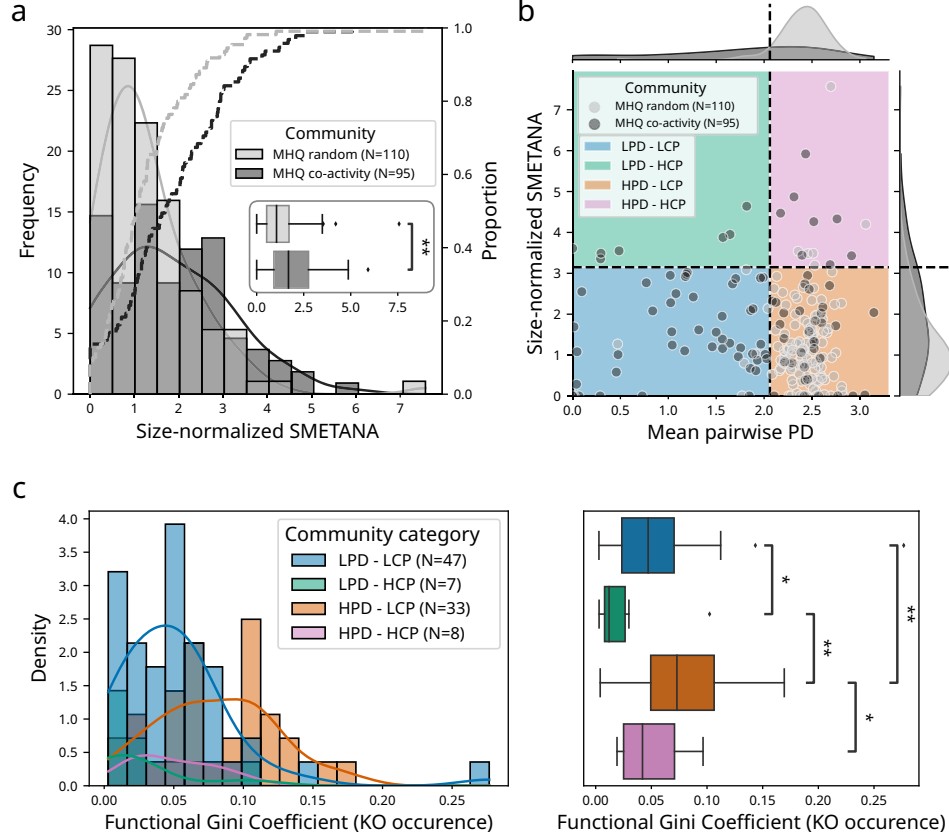

**Fig. 4 | Community-wide metabolic modelling reveals a higher metabolic interaction potential within marine prokaryotic communities. a** Microbial communities were delineated on the global co-active genome network using the MCL graph clustering algorithm (see "Methods"). Community metabolic modelling was performed using SMETANA on co-active communities ($N = 95$, dark grey, frequencies as bars and proportions as dashed line) and compared to random communities ($N = 110$, light grey). Boxplot insert: Co-active communities (dark grey) overall displayed a significantly higher metabolic interaction potential (SMETANA) score as compared with random communities (Mann–Whitney U test two-sided, $p = 1.09 \times 10^{-3}$). **b** Distinct metabolic interactions community types were identified within co-active marine prokaryotic communities (black points) and differentiated from random communities (grey points), the latter largely displaying an overall higher mean phylogenetic distance and lower metabolic cross-feeding potential score (HPD-LCP, orange quadrant): (i) Communities with overall low mean phylogenetic distance and low metabolic cross-feeding potential score (LPD-LCP, blue quadrant), (ii) communities with overall low mean phylogenetic distance and high metabolic cross-feeding potential score (LPD-HCP, green quadrant), and (iii)

communities with overall high mean phylogenetic distance and high metabolic cross-feeding potential score (HPD-HCP, pink quadrant). LPD co-active communities had a mean phylogenetic distance smaller than 95% of the random communities, while HCP have a mean SMETANA score above 95% of the random communities (dotted black lines). **c** HPD communities (orange $N = 33$, and pink $N = 8$) were more dissimilar to respective LPD communities (blue $N = 47$, and green $N = 7$) according to their functional Gini coefficient inferred from KEGG metabolism KO genes occurrence profiles (Mann–Whitney U test two-sided with Benjamini-Hochberg correction, LPD-LCP vs. LPD-HCP $p = 4.88 \times 10^{-2}$, LPD-HCP vs. HPD-LCP $p = 2.77 \times 10^{-3}$, HPD-LCP vs. HPD–HCP $p = 4.89 \times 10^{-2}$, LPD-LCP vs. HPD-LCP $p = 1.30 \times 10^{-3}$). The box extends from the lower to upper quartile values of the data (Q1 and Q3), with a line at the median (Q2). The whiskers extend from the box to show the range of the data and are defined as follows: where IQR is the interquartile range (Q3-Q1), the upper whisker will extend to last data point less than Q3 + 1.5 × IQR. Similarly, the lower whisker will extend to the first data point greater than Q1−1.5 × IQR. Beyond the whiskers, data are plotted as individual points.

enabling cooperative growth in scarce environments[56]. Such division of metabolic labour for AAs can promote a growth advantage for cross-feeding species, as the fitness cost of overproducing AAs has been experimentally shown to be less than the benefit of not having to produce them when they were provided by their partner[58]. Considering predicted L-glutamate exchanges, glutamic acids have been reported as potential auxophores (i.e., a compound that is required for growth by an auxotroph) in aquatic environments[59]. Notably, arginine and glutamate are linked in Cyanobacteria[60] and plants[61] through the metabolism of glutamate that involves the glutamate dehydrogenase for arginine synthesis, and which is an important network of nitrogen-metabolizing pathways for nitrogen assimilation. In marine micro-organisms, nitrogen (N) cost minimization is an important adaptive strategy under global N limitation in the surface ocean, acting as a strong selective pressure on protein atomic composition[62] and the structure of the genetic code[63]. Given that arginine plays an important role in the N cycle because it has the highest ratio of N to carbon

among all AAs, the combined selective pressure at genomic level and for biosynthetic (N) cost minimization may explain the recurrent cross-feeding predictions of glutamate and arginine observed herein. Overall, these results support amino acid auxotrophy as a potential evolutionary optimizing strategy to reduce biosynthetic burden under nutrient (in particular N) limitation while promoting cooperative interactions[56,64].

B-vitamins, which are essential micronutrients for marine plankton[65], are predicted here to significantly structure bacterioplankton community activity, which supports the hypothesis that B-vitamin mediated metabolic interdependencies contribute to shaping natural microbial communities[66]. A recent environmental genomic survey in estuarine, marine, and freshwater environments has revealed that most naturally occurring bacterioplankton are $B_1$ (thiamine) auxotrophs[67]. Vitamin interdependencies and auxotrophies, in particular for thiamine, have been recently predicted through a metagenomics-based association network in a soil microbial

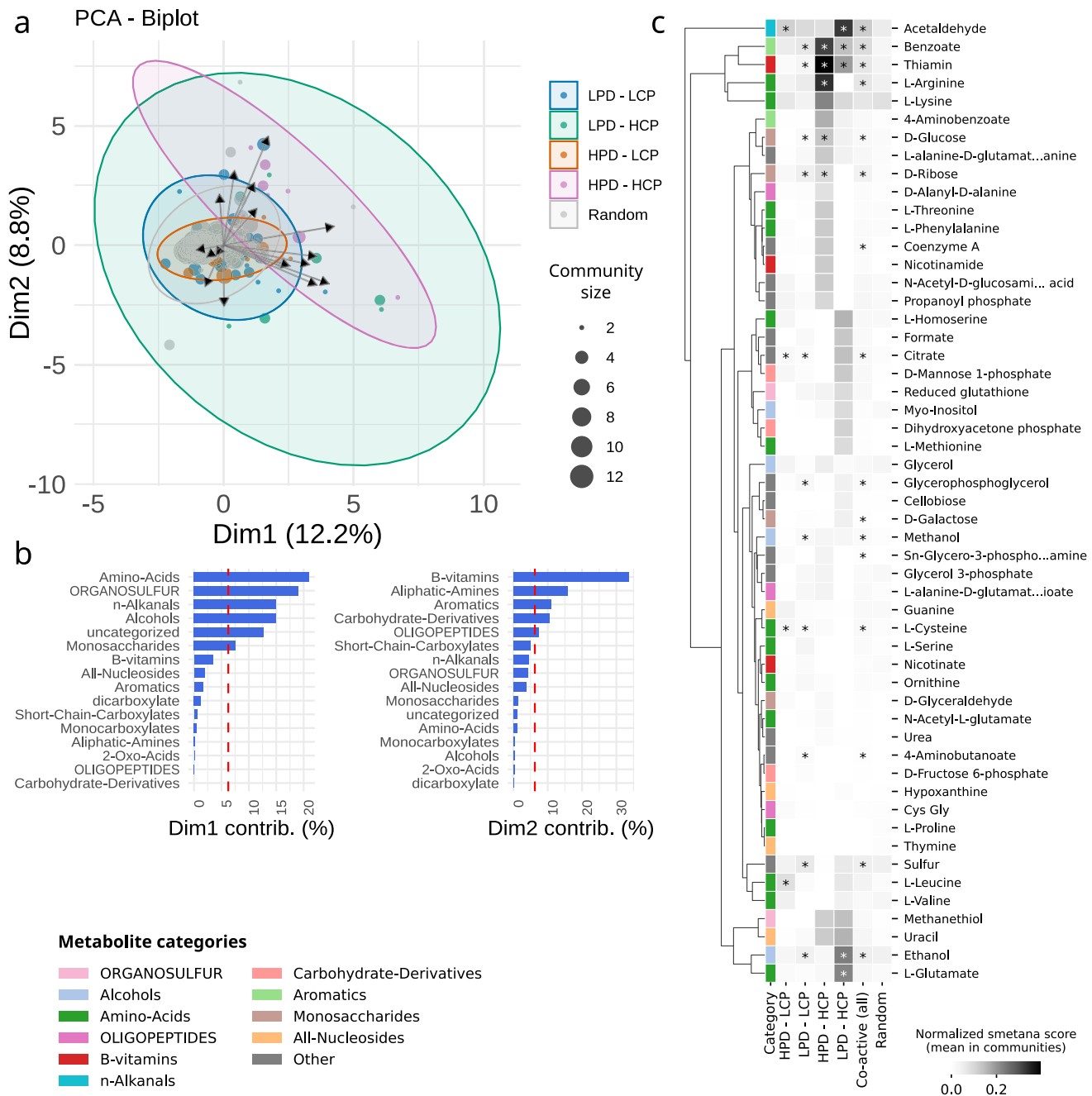

**Fig. 5 | Community metabolic modelling predicts specific metabolic cross-feedings within co-active marine prokaryotic communities. a** A NMDS analysis revealed that HPD-HCP and LPD-HCP communities are predicted to have a higher potential exchange in specific metabolic categories. Point size represents the size of each community in number of genomes. Coloured ellipses are visual aids to emphasize the distribution of points by categories. **b** Overall, the higher potential for exchanges in HPD-HCP and LPD-HCP communities is driven by specific metabolic categories (NMDS Dim1 and Dim2), in particular amino acids, B vitamins, organo-sulphur compounds, and aliphatic amines. **c** Within these broad metabolic categories, specific metabolite exchanges are identified within each co-active genome community type. Similar rows in the heatmap were clustered together by

hierarchical clustering (UPGMA algorithm). In particular, exchanges of acetaldehyde, benzoate, thiamin (vitamin B₁), ethanol, and L-glutamate are predicted in LPD-HCP, while in HPD-HCP exchanges of benzoate, thiamin, L-arginine, as well as D-glucose and D-ribose are predicted. Truncated metabolite names are (from top to bottom): L-alanine-D-glutamate-meso-2,6-diaminoheptanedioate-D-alanine, N-Acetyl-D-glucosamine(anhydrous)N-Acetylmuramic acid, Sn-Glycero-3-phosphoethanolamine, and L-alanine-D-glutamate-meso-2,6-diaminoheptanedioate. Stars denote a significant difference between categories (Mann–Whitney U, Benjamini-Hochberg correction, corrected $p$ value ≤ 0.05, all test results are available in Supplementary Data 8).

community, and confirmed in microcosm experiments[68]. Another comparative genomics assessment of vitamin B₁₂ (cobalamin) dependence and biosynthetic potential in >40,000 bacterial genomes predicted that 86% of them require the cofactor, while only 37% encode a complete biosynthetic potential, the others being split into partial

producers and salvagers[69]. In addition to thiamine, the joint importance in the metabolite exchanges of ornithine, glutamate and methionine, which are all products of enzymes dependent on vitamin B₁₂[70], confirms that access to vitamin B₁₂ plays a significant role in structuring microbial community interactions. Furthermore,

acetaldehydes are known intermediates supporting prokaryotic growth after breaking down substrates such as ethanolamine and propanediol using metabolic pathways involving vitamin $B_{12}$-dependent enzymes[71]. Taken together, our results thus support the prevalent reliance of bacterioplankton on exogenous $B_1$ and $B_{12}$ precursors/products and on the bioavailability of micronutrients as important factors influencing bacterioplankton growth and community assembly.

Given the identification of AAs, B vitamins and associated product exchanges as key metabolic mediators driving bacterioplankton community assemblies, we investigated the gene occurrence and activities of associated transporters in co-active communities (Supplementary Fig. 13). Transporter-associated reactions and associated genes were directly extracted from CarveMe models (using the gene-reaction rule), and classified into nine different types of transporter mechanisms, and two directions (import or export). This analysis revealed a vast majority of ABC systems for import reactions of specific AAs and B vitamins predicted exchanged, while more diverse transporter types (e.g., diffusion, proton antiport) were responsible for export reactions (Ext. Data Fig. 13a) across the four community types (Ext. Data Fig. 13b). Transporter gene activities (abundance-normalized expressions) confirmed and validated their transcriptional activity in genomes of the four types of co-active communities. Next, we also investigated the graph centrality of AAs and B vitamins donor and non-donor species within the co-activity network of bacterioplankton communities using the closeness centrality metric. The closeness centrality measures nodes centrality in a network by calculating the reciprocal of the sum of the length of the shortest paths between the node and all other nodes in the graph. The more central is a node, the closer it is to all other nodes. Overall, this revealed that donor species, in particular for AAs and B vitamins, displayed significantly higher closeness centrality than non-donor species (Supplementary Fig. 14). This observation supports the hypothesis that donor species may influence community assembly via cross-feeding interactions through more central positions or hubs in the ecological network. Given that metabolic interdependencies predicted here are mainly observed among co-active genomes that are overall smaller in size (Supplementary Fig. 6), we also compared the genome sizes of donor vs. non-donor species, which revealed that genomes of non-donor species tended to be significantly smaller in size as compared to genomes of donor species (Supplementary Fig. 14). This observation is in support of the Black Queen Hypothesis (BQH)[72,73], stating that species can gain a fitness advantage through genome streamlining, which is often observed (including herein) in free-living marine bacterioplankton genomes occurring in nutrient-limited ocean regions[74]. Genome streamlining can reduce the nutrient requirements associated with the maintenance of more genetic material and limits energetically costly metabolic activities. Here, genome streamlining and metabolic cross-feeding may act as joint mechanisms shaping free-living bacterioplankton community assembly in the oligotrophic surface ocean. Nonetheless, abiotic factors, such as temperature, are also likely conjointly impacting genome size of the surface ocean microbiome[75]. Although our prediction results underline the key role of metabolic cross-feeding supporting positive interactions between microbes, many microorganisms in nature are prototrophic and are able to grow on simple substrates without the help of others[76]. Also, some metabolic cross-feeding predicted herein may be due to substrate-based metabolic partitioning, allowing some community member to independently utilizes distinct substrates released or degraded from the same source[77]. Trade-off mechanisms such as resource allocation, design constraints, and information processing, can concomitantly shape microbial traits in the wild and lead to different biological adaptations leading to generalist or specialist lifestyles[78]. However, recent experimental work demonstrated that obligate cross-feeding can significantly expand the metabolic niche space of interacting bacterial populations[3], thus potentially positively selecting cross-feeding bacterial populations.

The metabolic cross-feedings and interdependencies predicted here can be extremely useful to draw hypotheses for testing in the laboratory, for example through co-culture experiments. Focusing on one of the most abundant photosynthetic organisms on Earth, the marine cyanobacteria *Prochlorococcus sp.*, we further analysed predicted exchanges within a small community of six genomes ('*coact-MHQ-014*', see Supplementary Data 7) including one genome of *Prochlorococcus marinus*, three genomes of Pelagibacteraceae (two *Pelagibacter sp.* and one *MED-G40 sp.*), one genome of order Rhodospirillales (family UBA3470), and one genome of phylum Dadabacteria (*TMED58 sp.*). The community biogeography of this consortium revealed a globally distributed activity in both SRF and DCM, but restrained to mainly Westerlies (temperate) stations between 30° to 60° in absolute latitude (mean 33.8°N/27.4°S in SRF, mean 34.3°N/21.7°S in DCM) (Supplementary Fig. 15). Most robustly predicted exchanges within this community included the exchanges of several amino acids (L-arginine, L-homoserine, L-lysine, and L-phenylalanine), of vitamin $B_1$ provided by a *Pelagibacter sp.* to two other genomes (*MED-G40 sp.* and family UBA3470), but also of D-ribose provided by the Rhodospirillales genome (family UBA3470) to *Prochlorococcus marinus*. The latter prediction provides a putative mechanism by which heterotrophic bacteria (such as from the order Rhodospirillales) can facilitate the growth of *Prochlorococcus marinus*[79]. While these metabolic exchanges remain predictions, they readily allow to formulate novel hypotheses to be further validated in the lab through co-culture experiments.

In sum, these results underline the global-scale importance of trophic interactions influencing the co-activity, assembly, and resulting community structure of marine bacterioplankton communities[2]. Our computational predictions support in particular amino acids and B vitamin auxotrophies[29,67] as likely important mechanisms driving bacterioplankton community assembly in the nutrient-limited surface ocean. Given that these metabolic interdependencies are mainly observed among co-active genomes that are overall smaller in size, these results support the Black Queen Hypothesis[72] as a potentially important mechanism shaping bacterioplankton community assembly in the global euphotic ocean. The integrated ecological and metabolic modelling framework developed herein has revealed the genomic underpinnings of predicted metabolic interdependencies shaping bacterioplankton community activity and assembly in the surface ocean. It also revealed putative trophic metabolic interactions occurring among the most abundant bacterioplankton cells in the ocean (i.e., *Prochlorococcus* and *Pelagibacter*). Ultimately, these in silico predictions will have to be validated experimentally, through (high-throughput) co-culturing[80]. Finally, the computational framework developed here can readily be applied to study other microbiomes, in which mechanistic predictions of biotic interactions may also serve for generating novel hypotheses for co-culturing, with the goal to better capture the vast uncultivated microbial majority across microbial ecosystems. Overall, this framework integrating ecosystem-scale meta-omics information through ecological and metabolic modelling paves the way towards an improved functional and mechanistic understanding of microbial interactions driving ecosystem functions in situ.

## Methods

### A database of species-level marine prokaryotic genomes
A database of genomes from marine prokaryotes was assembled using several specialised databases as well as genomes reconstructed within specific studies. These databases included whole-genome sequences from marine prokaryote isolates (WGS), single-amplified genomes (SAGs), and metagenomic-assembled genomes (MAGs). The main database source for our genome collection was the Marine

Metagenomic Portal[81] through the use of the databases MarRef v4.0 ($N = 943$, mostly high-quality WGS, available at https://mmp.sfb.uit.no/databases/marref/)[81], MarDB v4.0 ($N = 12,963$, available at https://mmp.sfb.uit.no/databases/mardb/)[81], and aquatic representative genomes from the ProGenomes database v1.0[32] ($N = 566$, available at http://progenomes1.embl.de/data/habitats/aquatic/aquatic.repr.contigs.fasta.gz). This collection of well-documented genomes was complemented by 5,319 MAGs assembled from four distinct studies, namely: Parks et al.[82] ($N = 1,765$; available at ENA under BioProject PRJNA348753), Tully et al.[83]/[20] ($N = 2597$; available at ENA under BioProject PRJNA385857), and Delmont et al.[21] ($N = 957$; available at https://doi.org/10.6084/m9.figshare.4902923). The Parks et al. study contained genomes reconstructed from non-marine biomes. Thus, a selection of 1,765 genomes was extracted by searching for specific keywords: "tara, marine, sea, ocean, mediterranean" (case insensitive). Note that depending on their study of origin, included MAGs may have been reconstructed using different assembling and binning methods. Details about included genomes and their origins are reported in Supplementary Data 1. Overall, our marine genomes catalogue contained 19,791 highly redundant genomes (WGS, MAGs and SAGs). Genomes from this non-dereplicated catalogue were further filtered and quality-controlled before their inclusion in our study. We used CheckM v1.0.18[84] to estimate the quality of the 19,791 genomes in our marine genomes catalogue (see SnakeCheckM in ecosysmic repository). Through the annotation and counting of single-copy marker genes (SCGs), CheckM estimates the level of completeness, contamination, and strain heterogeneity of individual genomes. We used those metrics to classify our genomes into three categories: high-quality (HQ) for ≥90% completeness and ≤5% contamination ($N = 8736$), medium-to-high-quality (MHQ) for ≥75% completeness and ≤10% contamination ($N = 4547$), and medium-quality (MQ) for ≥50% completeness and ≤25% contamination ($N = 5381$). Genomes that did not meet at least the MQ threshold were tagged as low-quality (LQ) and discarded from the database ($N = 1127$). Quality estimates were used in the de-replication process that was performed using dRep v2.2.3[85] (see dReplication in ecosysmic repository) using default parameters. dRep uses average nucleotide identity (ANI) and filters out redundant genomes via a 2-step clustering strategy: a fast coarse-grained clustering by MASH ANI (threshold used: 90% ANI over 60% of the genomes), followed by a slow fine-grained clustering through NUCMER ANI in clusters identified in the previous step only (threshold used: 95% ANI over 60% of the genomes). This process allowed to identify and select most complete and less contaminated genomes within each species cluster, which yielded 7658 non-redundant species-level genomes with an average nucleotide identity below 95%, a threshold previously reported to delineate species level for prokaryotes[86]. These genomes were assigned taxonomic information using GTDB-Tk v0.3.2[87] (see SnakeGTDBTk in ecosysmic repository), which also allowed us to place our genomes within a phylogenetic tree using iTOL v5[88]. Since GTDB-Tk reconstructs two independent trees for Archaea and Bacteria, we linked them at the root using a distance of 0.122[89], as recommended by the authors and tool maintainers (https://github.com/Ecogenomics/GTDBTk/issues/209).

## Functional annotations and reconstruction of genome-scale metabolic models

Coding DNA sequences (CDS) and proteins were inferred using Prodigal v2.6.3[90] and annotated using eggnog-mapper v1.0 on the eggNOG v5.0[91] orthology resource (see GeneAnnotation in ecosysmic repository). For genomic scaling laws analyses, annotated genes were regrouped into broad functional categories, using the Clusters of Orthologous Genes (COGs) 17 functional categories (e.g., replication, recombination and repair; nucleotide transport and metabolism), and the KEGG BRITE Functional Hierarchies (e.g., Energy metabolism; Metabolism of cofactors and vitamins). The sets of annotated genes

were processed using CarveMe v1.5.1[53] to reconstruct individual metabolic networks using the generic command "carve --output --universe --nogapfill --fbc2 --verbose " (see SnakeCarveMe in ecosysmic repository). The template used for each top-down reconstruction (referred to as "universe" in the original CarveMe paper) was selected for each genome using the GTDB-Tk taxonomic assignments as either cyanobacteria, bacteria, or archaea. CarveMe was run without gap-filling with the solver IBM CPLEX v12.10. The main reason for running CarveMe without gap-filling was to avoid predicting potential false positive cross-feeding metabolic interactions. Given the uncomplete nature of genomes we used, this is a conservative approach, as without gap-filling we also likely miss potential true cross-feeding metabolic interactions.

## Genomic scaling laws analysis

Since genome size was spanning between several order of magnitudes in our database, we had to account for its effect when comparing the functional content of specific groups of genomes (e.g., origin, co-active or not). For a given functional category, such a relationship can be modelled by a so-called scaling law[36], a power law that links the number of genes in the category with the total number of genes in the prokaryotic genome. By identifying deviations from the power law (or linear law in log-log scale), we used scaling laws as a tool to properly identify enriched or depleted functional and metabolic potentials within groups of genomes in our genomic database. In other words, to identify if a genome or group of genomes harbours fewer or more genes in a category than what would be expected given its size. Egg-NOG provides 25 high-level categories and a KEGG Orthology (KO) equivalent for each Cluster of Orthologous Group (COG) annotation. The KO database also provides a 4-level hierarchy of (unnamed) functional categories. We were able to group our 23,224 KO identified in our catalogue into 54 high-level categories (level 2 in the hierarchy that presented for us the best compromise between specificity and tractability of the metabolic functions). For each high-level KO or COG category, we fitted a linear law on the log-transformed variables using the function scipy.stats.linregress v1.7.3 (parameter alternative = " greater"). Functional categories with a $R^2$ below 0.3 were discarded, and the distribution of residuals were compared (in log-scale) using the Mann-Whitney U test using the function scipy.stats.mannwhitneyu v1.7.3 (parameter alternative = "two-sided"). $P$ values from all tests were corrected using Bonferroni and Benjamini-Hochberg multiple-testing corrections (see Supplementary Data 3, 5, and 8) using the function stats.multitest.multipletests from the statsmodels Python package (v0.13.2).

## Functional distances and Gini coefficient

We defined the functional relatedness between genomes by computing KO-based functional Jaccard distances between genome vectors of KO gene presence/absence. In addition, we computed a complementary functional distance based on gene expression inferred from metatranscriptomics data. Similarly, KO-based functional Jaccard distances between genome vectors of KO expressed gene were used, with a KO being expressed when its expression was detected in at least 10 samples.

To quantify how the functional potential of each community was shared between genomes, we used a proxy of the well-established Gini index. In Economics, the Gini index "measures the extent to which the distribution of income (or, in some cases, consumption expenditure) among individuals or households within an economy deviates from a perfectly equal distribution". Inside each predicted co-active consortium, we defined a "functional capital" for each member as the sum of occurring KO that were present inside the genome, and computed the Gini index on this value. A Gini index of 0 can be interpreted as a perfect overlap between the functions of all members of the consortium, while a Gini index of 1 would be the extreme situation where a

single member of the consortium displays all the detected KO functions. Intermediate values represent varying degree of metabolic evenness between the members of the community, a measure that we tried to use to separate niche overlap from potential metabolic complementarity.

## Meta-omics profiling and associated environmental contextual data

We leveraged metagenomics and metatranscriptomics data from samples of the *Tara* Oceans expeditions (2009–2013)[92]. We focused on samples from prokaryotic-enriched size fractions (0.2–1.6 μm and 0.22–3 μm) in the euphotic zone, including surface (SUR) and deep-chlorophyll maximum layer (DCM) samples. This yielded 107 samples across 64 stations for metagenomics data, 118 samples across 81 stations for metatranscriptomics data, and 71 samples across 45 stations for which we had both. Prior to metatranscriptomics sequencing, cDNA synthesis of total RNAs was performed by a random priming approach preceded by a prokaryotic rRNA depletion step[92]. Sequencing reads were previously quality-controlled using methods described in[92]. We then mapped quality-controlled reads onto our 7,658 non-redundant marine prokaryotic genomes using Bowtie 2 v2.3.4.3[93] (see ReadMapping in ecosysmic repository) using the command "bowtie2 -p --no-unal -x -1 -2 -S" with no extra parameter. Reads that successfully mapped were subsequently filtered using Samtools v1.9[94] and pySAM v0.15.2 using MAPQ ≥ 20 and a nucleotide identity ≥ 95% to avoid non-specific mappings. The identity score ignores ambiguous bases (N) on the reference but takes gaps into account. The formula used is (NM - XN) / L with NM the edit distance; that is, the minimal number of one-nucleotide edits (substitutions, insertions and deletions) needed to transform the read string into the reference string, XN the number of ambiguous (N) bases in the reference, and L the length of the read. Overall, this ensured that the conserved reads were mapped to the target genome with a high-specificity. The presence, abundance, and activity of a given genome was determined as follows: first, the occurrence of a genome was determined by its horizontal metagenomic coverage of minimum 30%; second, its abundance was computed using its vertical metagenomic coverage normalised by its genome length. Finally, its activity corresponded to the ratio of its vertical metatranscriptomic coverage over its vertical metagenomic coverage, and no genome activity threshold was set, thus potentially resulting in null activity for some genomes in some samples. We estimated depth of coverage (i.e., vertical coverage) by dividing the total mapping of a genome by its size, and breadth of coverage (i.e., horizontal coverage) by dividing the number of mapped bases (at least one time) by the genome size (see CoverageEstimation in ecosysmic repository).

## Co-abundance and co-activity networks inference

Genome-resolved co-abundance and co-activity networks were reconstructed using *FlashWeave* (FW) v0.18.0[50]. FW first infers a global correlation network resulting from significant partial correlation tests with Fisher's z-transformation ("sensitive" mode) between genome pairs co-abundance or co-activity profiles. Next, FW relies on a local-to-global learning framework that implements conditional independence tests to detect and remove indirect associations within this global network. Several heuristics are then applied to connect these local dependencies and infer a network. Finally, the resulting network integrates only direct associations, which edges are weighted by the partial correlation strength. Starting from all MQ genomes (*N* = 7658), we defined the abundance of a genome in a sample by its overall metagenomic vertical coverage (also called depth) per 1 M base pairs, while its activity was given by the ratio of its overall metatranscriptomic coverage depth per 1 M base pairs over its abundance. Note that this can only be computed at stations and depths for which we have both metagenomic and metatranscriptomic signals. A given genome was defined as observed

(i.e., present and/or active) within a sample when at least 30% of its genome was horizontally covered (also called breadth).

Overall, we were able to compute abundances for 107 samples, and activities for only 71 samples. To lower spurious correlations, abundance and activity data points for unobserved genomes were discarded and genomes with less than 10 observations across our samples were removed. This was done independently for abundance (*N* = 1232 genomes observed in at least 10/71 samples) and activity (*N* = 902 genomes active in at least 10/71 samples). Finally, the inherent compositional nature of the sequencing datasets was taken into account using centred log-ratio (CLR) transformation and the adaptive pseudo-count implemented in *FlashWeave*. Both abundance and activity matrices were used as input to *FlashWeave* using parameters "normalize=true, "n_obs_min=10, max_k = 3, heterogenous=true" (see the FlashWeave documentation for more information about these parameters). Genome graph centralities were computed with the *networkx* python library v3.1 using the *closeness_centrality* function on the co-activity community networks for which metabolic exchanges were predicted using SMETANA (see below).

## Community metabolic modelling and cross-feeding interaction predictions

We identified co-active genome communities in the reconstructed co-activity network using the Markov clustering algorithm[95] (MCL) through the use of *run_mcl* function with an inflation parameter of 1.5 available in Python *markov_clustering* library V.0.0.2. We also generated randomly-assembled communities by randomly sampling genomes from the pool of genomes used for network reconstruction (genomes occurring at least 10 times within the considered samples). These communities were quality-filtered for MHQ genomes and analysed using SMETANA 1.2.0[55] to predict putative metabolic cross-feeding interactions (see SnakeMETANA in ecosysmic repository). SMETANA does not use any biological objective functions and is formulated as a mixed linear integer problem (MILP) that enumerates the set of essential metabolic exchanges within a community with non-zero growth of all community species subject to mass balance constraints. Here, SMETANA was used to compute three distinct metabolic inter-action scores: (i) the Metabolic Resource Overlap (community-wide) score (MRO) quantifies how much species in a given community compete for the same metabolites, (ii) the Metabolic Interaction Potential (community-wide) score (MIP) calculates how many metabolites a given species can share to decrease their dependency on external resources, and (iii) the SMETANA score (pairwise between two species) that evaluates the probability of a cross-feeding interaction (two species, one direction, one metabolite) by integrating three additional metrics: (a) the SCS (species coupling score), which measures the dependency of one species in the presence of the others to survive, (b) the MUS (metabolite uptake score), which measures how frequently a species needs to uptake a metabolite to survive, and (c) the MPS (metabolite production score), which measures the ability of a species to produce a metabolite. The SMETANA sum score (sum of the SMETANA scores in a given community) is employed by the original authors and in our study as a community-wide version of the pairwise SMETANA score. We limited the community metabolic analyses to MHQ genomes in order to lower the risk of predicting spurious interactions in communities of lower-quality genomes and metabolic models. SMETANA was run in both global and detailed modes with the solver IBM CPLEX v12.10, using in each mode the default media provided by the package (which is a complete media for global analysis, and a community-specific minimal media for detailed analysis). A set of inorganic compounds were excluded from the analysis as explicitly recommended by one of the package author (https://github.com/cdanielmachado/smetana/issues/20#issuecomment-827389107). Other parameters used were "--flavour bigg --solver CPLEX --molweight".

The "community smetana score" reported in the main text is obtained by summing all smetana scores predicted for a given community. In order to compare communities of different sizes, this score was normalised by dividing the "smetana score" by the total number of potential genome-genome interactions, i.e. N x (N-1) / 2 (with N the size of the community). We referred to this new score in the main text as "normalised smetana score".

In order to classify the different metabolites in the SMETANA database into metabolite categories (e.g., amino acids, carboxylates), we first mapped the metabolite identifiers to the MetaNetX database (available at: https://www.metanetx.org/cgi-bin/mnxget/mnxref/chem_xref.tsv). From this mapping, we extracted MetaCyc identifiers to subsequently obtain their ontologies (available at: https://metacyc.org/groups/export?id=biocyc14-14708-3818508891&tsv-type=FRAMES). In this process, a number of metabolites could not be assigned to any metabolite category and were dumped as "uncategorized".

### Transporters activity

Transporter-associated reactions were directly extracted from CarveMe reconstructed models and classified into nine different types of transporter mechanisms and two directions (import or export). Reversible transport reactions were duplicated and counted in both directions. Gene activities (abundance-normalized expressions) were obtained by exploiting the so-called gene-reaction rule encoded within the CarveMe models. These rules often involved the expression of multiple genes to have a functional transporter protein, but since metagenomics and metatranscriptomics data are known to be sparse (especially at gene level), we considered a transporter active if at least one of its components was actively transcribed. Overall, we identified AAs and B vitamins transporters and associated reactions, linked them with actual genes present in co-active genomes, and detect their transcriptional activities across co-active community types (Supplementary Fig. 13).

### Statistical analyses

All statistical tests and analyses were performed using *scipy.stats* Python module v1.7.3. All figures were generated using Python v3.7.12 and R v4.2.2. We used statannotations v0.4.4 (https://github.com/trevismd/statannotations) to append statistical significance to all boxplots. Stars are used to define significance level as follow: **** for $p \leq 10^{-4}$, *** for $10^{-4} < p \leq 10^{-3}$, ** for $10^{-3} < p \leq 10^{-2}$, * for $10^{-2} < p \leq 5 \times 10^{-2}$, and finally ns for $p > 5 \times 10^{-2}$. All data analysis sub-packages were installed in the same environment using Conda v22.11.1, the versions of which are detailed in the yaml file located in each repository cited above.

### Reporting summary

Further information on research design is available in the Nature Portfolio Reporting Summary linked to this article.

## Data availability

The main database source for our genome collection was the Marine Metagenomic Portal through the use of the databases MarRef v4.0 ($N = 943$, available at https://mmp.sfb.uit.no/databases/marref/), MarDB v4.0 (N = 12,963, available at https://mmp.sfb.uit.no/databases/mardb/), and aquatic representative genomes from the ProGenomes database v1.0 ($N = 566$, available at http://progenomes1.embl.de/data/habitats/aquatic/aquatic.repr.contigs.fasta.gz). This collection of genomes was complemented by 5319 MAGs assembled from four distinct studies, namely: Parks et al. 2017 ($N = 1765$; available at ENA under BioProject PRJNA348753), Tully et al. 2017/2018 ($N = 2597$; available at ENA under BioProject PRJNA385857), and Delmont et al. ($N = 957$; available at https://doi.org/10.6084/m9.figshare.4902923). All *Tara* Oceans metagenomes and metatranscriptomes raw reads are available at ENA under BioProject PRJEB402. All data associated with the analyses are available in the supplementary materials and at Zenodo: https://zenodo.org/record/7853699#.ZEQ8ahVBx0Q.

## Code availability

All code repositories cited below are available within https://gitlab.univ-nantes.fr/ecosysmic.

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

## Acknowledgements

Tara Oceans (which includes both the Tara Oceans and Tara Oceans Polar Circle expeditions) would not exist without the leadership of the Tara Ocean Foundation and the continuous support of 23 institutes (http://oceans.taraexpeditions.org). We wish to thank the commitment of the following sponsors: CNRS (in particular Groupement de Recherche GDR3280 and the Research Federation for the study of Global Ocean Systems Ecology and Evolution, FR2022/Tara Oceans-GOSEE), European Molecular Biology Laboratory (EMBL), Genoscope/CEA, The French Ministry of Research, and the French Government 'Investissements d'Avenir' programmes OCEANOMICS (ANR-11-BTBR-0008), FRANCE GENOMIQUE (ANR-10-INBS-09-08), the CNRS MITI through the interdisciplinary program Modélisation du Vivant (GOBITMAP grant to SC), the RFI ATLANS-TIC2020 (ECOSYSMIC grant to SC), and the H2020 project AtlantECO (award number 862923). NG and ED were supported by the RFI ATLANSTIC2020 (ECOSYSMIC and PROBIOSTIC grants). We also thank the support and commitment of Agnès b. and Etienne Bourgois, the Prince Albert II de Monaco Foundation, the Veolia Foundation, Region Bretagne, Lorient Agglomeration, Serge Ferrari, World Courier, and KAUST. The global sampling effort was enabled by countless scientists and crew who sampled aboard the Tara from 2009–2013, and we thank MERCATOR-CORIOLIS and ACRI-ST for providing daily satellite data during the expedition. We are also grateful to the countries who graciously granted sampling permissions. Computational support was provided by the bioinformatics core facility of Nantes (BiRD - Biogenouest), Nantes Université, France. The authors declare that all data reported herein are fully and freely available from the date of publication, with no restrictions, and that all of the analyses, publications, and ownership of data are free from legal entanglement or restriction by the various nations whose waters the Tara Oceans expeditions sampled in. This article is contribution number 150 of Tara Oceans.

## Author contributions

S.C. designed the research. N.G., M.G., and S.C. analysed the data, performed bioinformatic analyses, and analysed the results. C.T. and E.D. analysed the data and performed bioinformatic analyses. N.G., M.G., C.N., C.B. and S.C. interpreted the results. N.G. and S.C. wrote the paper with inputs from all other authors.

## Competing interests

The authors declare no competing interests.

## Additional information

**Peer review information** : *Nature Communications* thanks Shengwei Hou and the other, anonymous, reviewer(s) for their contribution to the peer review of this work. A peer review file is available.

