## [Peer Review File · Nature Communications]

Genome-scale community modelling reveals conserved metabolic cross-feedings in epipelagic bacterioplankton communitiesReviewer #1 (Remarks to the Author):

The article entitled "Genome-scale community modelling reveals conserved metabolic cross-feedings in epipelagic bacterioplankton communities" by N. Giordano and coauthors reports the analyses of data generated by the Tara expeditions in order to predict metabolic interactions (cross feeding and competition for organic resources) between active bacterial populations detected in the deep chlorophyll maximum zone or the surface. This work is one great example of ecosystems biology promises and generated data can certainly allow experimental validation of predicted interactions. Figures are generally very appealing and the partitioning between the main figures and the extended data figures is essentially convincing. Overall, this article reads well albeit some information is missing from the main text to facilitate the reader understanding. Please find below some comments and suggestions.

Major comments

1-Genomes quality thresholds is an upstream key aspect of the study, and necessitate some clarifications:

- a) Does MHQ MAGs contain HQ MAGs? This is indeed what I understand from the given description of MHQ MAGs (Line 101: ">75% complete with less than 10% contamination" contains genome included in ">90% complete with less than 5% contamination"), but Figures shows separated set of genomes. In addition, line 142, MHQ are now defined as " $\leq 5\%$ " contaminated. The same issue rise with the definition of MQ MAGs, which could contain MHQ and HQ. As the selection of reliable genome is the foundation of all the following work, it would be useful and more convincing to have clearer definitions of the different genome subsets.
- b) Line 116: "we limited our analysis to HQ and MHQ genomes, which were of equivalently high-quality": They are not of equivalently high-quality since the used quality thresholds are not the same. However, the scaling law residuals distributions are not significantly different for HQ and MQ (but is different of MHQ...).
- c) Mainly for MAGs, plasmids and other genetically mobile elements are generally not recovered during binning. What are the consequences for your analyses? Can you exclude this as a confounding factor when analyzing the genome sizes and CDS numbers (line 118-120: "This analysis also [...] with WGS genomes.")? Is it possible to compare MAGs to WGS depleted from their plasmid-encoding contigs and other mobile elements?
- d) From Line 275 on, which quality of genomes is used? MHQ?
- e) Line 487-490: during dereplication, which genome replicates is conserved? Usually it is the most complete and less contaminated, but this could have been adjusted by the user.
- f) Genome networks were computed on which genome quality subset?

2- At some instances, adding a short sentence or two explaining the strategies, the methods or the content of already published work would ease the reading, so that the reader does not need to scroll continuously to the methods section or find the reference to get the information necessary to understand or get convinced by the analyses.

- a) A broad readership will maybe not know how to read the residuals plots. A short explanation is probably necessary.
- b) Can you provide some information of the sequencing, albeit being already published? I was notably wondering about the sequencing depth in relation to the saturation, mainly for metatranscriptomics. Is it equivalent for all samples? If not, how does it impact the active sub-community analysis? As the more abundant genomes are also the more active ones, can you provide a short explanation on if the RNA was treated not to be DNA-contaminated to rule out this confounding factor?
- c) Line 196: Albeit pointing to the method here, giving the 'activity threshold' would make the concept/strategy clearer.
- d) Line 251, SMETANA is introduced. I still did not get how were computed MRO and MIP scores from the SMETANA score, or even if it's the contrary.
- e) CarveMe was run without the time and resource consuming gap filling step. What are the

consequence for the cross feeding model and analyses? Can this be discussed?

3-Displays and their legend: albeit Figures are very eye-catching, some details could be modified to make them also really self-explanatory. Please find a list of instances where small modifications could be helpful to the readers.

a) Caption of the main figures and extended figures are sometimes incomplete:

Fig 2B: what is "hor.cov"; The red dotted line is generally not described (cryptic in Fig 2b, and absent in Fig 3, extended Fig 1, 2 and 5). It is described in Extended Fig 7, but to depict something different; In figure 3, the color code is confusing, with a color gradient that is not associate to the 3 colors used in the violin plot and in the legend. Where does stop the red and starts the grey? Please correct "N=?"; In figure 4, what does represent the insert? (residuals, as for the previous figures?). In panel b, how were the thresholds select? Please be consistent with the utilization of the abbreviation "PD" rather that "phyl.distance"; In figure 5: please give the stress of the NMDS in panel a. Please, expand the legend: How were the ellipses placed? What is the symbol size range from 2 to 12 indicating? What is the hierarchical clustering based on? What is the meaning of stars? Please homogenous the metabolite naming, notably for LalaDgluMdapDala, LalaDgluMdap, g3pe, Cys Glu, anhg, etc... This comment also applies to the related Extended Figure 10.

b) Extended Tables have no title, and some columns headers are not easy to understand. There is two Extended table 1 and two Extended table 2, which is pretty confusing.

In Extend data Fig.1, panel d displays a reduced number of genome compared to other panels. Can you explain? In Extend data Fig 5, please explain $d > 2$ or $d < 2$; In Extend Fig 5 and 6, explain in the legend what are the photic samples (in grey); In extend figure 7 and 8: "size" of what?; Please provide the y-axis unit in Extended data figure 9 and 11.

4-Networks, notably their construction, are not enough explained. Which type of correlations were computed? Were edges weighted (the correlation strength)? How can you show us that "This genome-resolved co-activity network was significantly different than the corresponding genome-resolved co-abundance network" (line204-5). Can you discuss the presence or absence of (disconnected) modules in these networks? How variable is the activity network if the threshold of activity is moved away from 30%?

5-General questions on the presented results:

a) The discussion from Line 120 to Line 126 is not clear to me. How did you select the 4 KEGG categories out of the Extended Table 2? A lot more categories seems to meet the statistical objectives. Why focusing on KEGG rather than COG? What about displaying a dotplot for every category so the reader can check out?

b) Line 216-219, you discuss the potential niche overlap (and thus possible competition) between short PD taxon. As you have the transcriptomic data for them, can you verify your assertion checking which genes are expressed and if they cover the same function? It would be much more convincing.

c) In co-active communities and from the metabolic reconstruction, what is the proportion of 'selfish' taxa and of 'altruistic' or 'generous' taxa? Is there some trends?

d) Are the HCP community described around line 326-330 sharing some environmental specificities? Are they derived from region with a high or a low productivity? Are some physico-chemical parameters specific to the related sampling area?

e) Genomes and active genomes observed in less than 10 samples were removed. Have you check their depth of coverage? Are you removing only low abundance / low activity taxon? Or are you also removing taxon with local high abundance/activity?

Minor comments

1 - Line 102 "HQ and MHQ MAGs were not significantly different from WGS genomes in terms of gene density (Supp. Table 1)": I did not find the statistic for this assertion in the mentioned display.

2 - Line 107: what is a "high-level functional category"?

3 - Line 118: please correct the typo "...that HQ/MHQ MAGs and were systematically..."

- 4 - Line 124 "uncultivated genomes": please rephrase, as no genomes are cultivable yet ;-)
- 5 - Line 145 chlorophyl versus chlorophyll
- 6 - In Figure 2B, the titles "abundance" and "activity" looks like axes titles. Can they be slightly moved?
- 7 - Line 210 "The co-activity network revealed a larger number of significant positive associations across large phylogenetic distances (PD), while negative associations were mainly observed between phylogenetically distant genomes ...": Isn't it contradictory? From the figure, I understood that positive associations are rather between population of short phylogenetic distances.
- 8 - Line 222 cites Fig.1, but this figure does not show the abundance of genome versus their size.
- 9 - Please tone down Line 253-4: "... to predict metabolic interaction potential and reveal [possible] metabolic exchanges and cross-feedings..."
- 10- Line 257: what does mean "community species"
- 11- Line 267 "...which we thus normalised by community size ...": is it meant co-active communities or the total community?
- 12 - Last sentence of the paragraph 260-274, and the first sentence of the following section (line 275-277) reads contradictory. Or should "large" line 275 rather reads divers/various?
- 13- Line 275-6 "...among co-active genomes (Fig. 3a) and communities...": which community are you referring to?
- 14 - Line 278-9 "largely composed of genome communities": please rephrase, what are 'genome communities'?
- 15 - Line 317-8 "Both HPD-HCP and LPD-HCP communities were predicted to have a higher potential exchange in specific metabolites as revealed by a NMDS ...": isn't it the definition of HCP (high cross feeding potential)?
- 16 - line 319 "large metabolic categories": do you mean broad?
- 17 - Line 382 "we investigated their graph centrality...": who is 'their'. It reads like if it was metabolites.
- 18 - Line 386 and line 387 "genome donors" and "non-donor genome": please rephrase, it is not genome that encode the metabolic potential of sharing, and not the genome that is providing the metabolite
- 19-Line 492 "GTDB-TK": please check for the caps.
- 20 - Line 572 "... and activity (N=902 genomes observed in at least 10/71 samples)." should read "... and activity (N=902 *active* genomes observed in at least 10/71 samples)."
- 21 - Line 929-930 "We used samples with both metagenomics and metatranscriptomics available to compute genome-wide co-activity.": can this information be in the main text?
- 22 - Line 949-950 "Genomes in the co-activity network are significantly smaller both in size and number of CDS": I don't see that.

Reviewer #2 (Remarks to the Author):

This is an exciting study that takes a significant step beyond the ubiquitous abundance correlation networks so common in microbial ecology literature. By leveraging information from extensive genome database and environmental omics data sets, the authors provide a roadmap for functional and metabolic analysis. They aim to expand upon co-occurrence networks by utilizing genome-resolved metagenomics data sets together with meta-transcriptomic data to uncover activity in transcription. The authors rely on genome scaling laws to characterize the functional content of coactive genomes, identifying functional gene categories that might drive metabolic dependencies in the community. By employing genome metabolic models, they uncover potential cross-feeding interactions

This study addresses an important problem in marine microbial ecology, specifically how microbial communities are assembled depending on their functional capabilities. The authors are particularly interested in developing a mechanistic understanding of how metabolic auxotrophy constrains community composition and assembly. This is an exciting study that does much to go beyond the

abundance correlation networks that are so ubiquitous in microbial ecology literature now. It does this by starting to leverage the genomic information in our genome database and environmental omics data sets. The field will likely find this manuscript useful in providing a road map to do such functional and metabolic analysis.

Major comments

I completely agree that functional, metabolic and ecosystem modeling approaches, like the one taken here, are needed to understand the complex interactions and emergent properties of these microbial communities from the genome content and how they fit into ecological and biogeochemical roles. However, my main concerns with the manuscript are its sheer density and the fact that the authors have packed a lot of information into a relatively small space, making the logic sometimes difficult to follow. This issue seems to stem from having to bounce between results and methods or potentially descriptions in the figures. The manuscript would likely be more readable if a figure guided the reader through the progression of the different analyses and metrics used.

My other major concern was the choice to use an external genome data set to map the TARA omics data to, as the percentage of TARA reads mapped to this database was relatively low. This raises a methodological concern about the choice to use the genome catalog versus trying to use assembled genomes directly from the Tara oceans data set. It's not clear why this decision was made. I would expect if you would have used the assembled genomes from Tara you would have had a higher MG and Mt read mapping. The authors should provide an explanation for why they chose to go with the external genomes versus the Tara assembled genomes and clarify this in the text.

Overall, my suggestion is for revisions to increase the clarity and digestibility of the manuscript, particularly given the density of analysis conducted. This will ensure that this important work can be more accessible to the many who will be interested in it.

Line 508: "We used scaling laws as a framework to characterise the functional content of our genomic database"... This is such a key piece of the analysis conducted, but it's not very clear what it actually means. The genomes have their functional content based on their genes and resulting KEGG and eggnoG annotations. What and how were the scaling laws used to do. Apologies if this is something that should be obvious, but I was having a hard time making the specific connections from the way its described in the manuscript.

Detailed Comments

Line 118: typo, "and" at the end of the line.

Line 119. Is this observation supposed to be able to be seen in figure 1B? If so it seems the MAG span the genome sizes in the figure.

Line 122 - 126: There is a lot here. It seems like an important part of the analysis but it's only briefly described how this was done, with most of it in the supplementary information. Would it be helpful to expand on this a bit to help clarify to the reader what's going on. Also the direction of the trend is not specifically mentioned. For example, I assume that decrease metabolic potential for xenobiotics and polyketone aids is in smaller genomes, but this is not explicitly stated.

Line 155: There is a lot in the results that the authors force the reader to bounce back and forth between the methods and results. For example the genome-wide activities. It would be much easier to read and digest the study if they could include a simple description of what the genome-wide activity metric is when it is first mentioned. Or how what the co-activity to co-abundance comparison metric is.

Line 157. What is the metric that you are using for the PCA in terms of species abundance? Coverage of mapped reads to the genomes? I appreciate that there are lots of analyses in this study and the importance of being concise, but often this makes it difficult to follow exactly what is happening.

Fig 2. Does the differences in sample coverage between the MG and MT datasets lead to biases? It's a bit hard to tell, but it looks like there is less of North Atlantic and Mediterranean samples in the MT versus the MG. This could cause biases in the environmental factors identified as correlating with the PCA axis.

Line 195: IS this activity different than the one mentioned above?

Line 206: This could use clarification. Do you mean a small fraction shared between the co-activity and co-abundance networks?

Line 288: The gini coefficient is first mentioned here but not explained what it means until the methods.

Line 330 typo "quote zooming in on"

Line 339 it's not clear the distinction between these two community comparisons what the difference is between "enriched" and "predicted".

The example with *Prochlorococcus* and other taxa was extremely useful. Could this be expanded for other organisms in your analysis. This section likely provides the most concrete take away takeaways from the entire manuscript.

Figures sometimes need more detailed explanations especially since they are so dense. For example figure 5a: what are the circle sizes supposed to represent? Or in figure 5B what do the asterisks mean?

Reviewer #3 (Remarks to the Author):

Identifying how active microbes interact with each other in a community is essential to predict microbial community composition, which has a crucial impact on the element cycling and primary production of the oceans. Using Tara Oceans meta-omics data, Giordano et al predicted inter-lineage associations between co-active microbial communities of the euphotic ocean, which was interpreted as conserved cross-feedings of cellular metabolites, particularly costly amino acids and group B vitamins. In addition, the authors also found that microbes with high cross-feeding potentials were overall smaller in genome size, thus proposing that genome streamlining and metabolic auxotrophies were central joint mechanisms shaping the assembly of bacterioplankton communities in the epipelagic ocean.

Overall, this is a high-quality manuscript with sufficient novelty and broad community interest. The content is well organized with an adequate introduction, detailed method description, accurate result interpretation, insightful discussion, and appropriate references. However, as currently submitted, there are two major concerns that should be carefully addressed.

Firstly, the authors acknowledged that the prediction of metabolic exchange within co-active communities should be further validated in the lab through co-culture experiments, which I believe is well beyond the scope of the current manuscript. However, the current prediction is not convincing without further analysis of the exchange mechanisms between donors and receivers/non-donors. I kindly request the authors investigate the transporters responsible for specific amino acids and group

B vitamins in co-active communities. It would be beneficial to confirm if these transporters are present/enriched in the non-donor genomes and actively transcribed. It is possible that some of the observed cross-feeding is due to substrate-based metabolic partitions where each community member independently utilizes distinct substrates released or degraded from the same source.

Secondly, the claim that genome streamlining and metabolic auxotrophies were the central mechanisms shaping the assembly of microbial communities in the surface ocean may be overstated. Black Queen (BQ) processes may lead to mutualistic interactions and genome reduction of non-donors if they have sufficiently large effective population sizes (N_e), otherwise, BQ gene loss and genetic drift may ultimately lead to obligate symbiosis (Giovannoni et al., 2014 ISME J). Since all the non-donors discussed in this study are free-living, they either have a large N_e or are only losing costly traits instead of undergoing significant streamlining evolution. In addition, genome streamlining is more prevalent in nutrient-limited environments, suggesting abiotic factors shouldn't be ignored in the discussion.

Minor comments:

1. L118 and L142, has the genome size been normalized by its completeness?
2. L156, have rRNA reads been depleted or not?
3. Fig 2c, it would be better to have the regression lines in the subplots.
4. L215, how do you define "functionally related"?
5. Fig 5a, what does the point size mean?
6. Fig 5b, what's inside of "uncategorized"?
7. "p" should be italicized in "p-value" at multiple locations.
8. Similarly, the multiply symbol should be used instead of "x" at multiple locations.

REVIEWER COMMENTS

Reviewer #1 (Remarks to the Author):

The article entitled “Genome-scale community modelling reveals conserved metabolic cross-feedings in epipelagic bacterioplankton communities” by N. Giordano and coauthors reports the analyses of data generated by the Tara expeditions in order to predict metabolic interactions (cross feeding and competition for organic resources) between active bacterial populations detected in the deep chlorophyll maximum zone or the surface. This work is one great example of ecosystems biology promises and generated data can certainly allow experimental validation of predicted interactions. Figures are generally very appealing and the partitioning between the main figures and the extended data figures is essentially convincing. Overall, this article reads well albeit some information is missing from the main text to facilitate the reader's understanding.

Please find below some comments and suggestions.

We wish to thank reviewer 1 for his useful and constructive comments, and appreciate the feedback about the relevance of our ecosystems biology approach and associated findings on predicted metabolic interactions in upper-ocean bacterioplankton communities. Below we detail specific actions we have taken to address his comments. All modifications are **highlighted in yellow** in the revised manuscript.

Major comments

1-Genomes quality thresholds is an upstream key aspect of the study, and necessitate some clarifications:

a) Does MHQ MAGs contain HQ MAGs? This is indeed what I understand from the given description of MHQ MAGs (Line 101: “>75% complete with less than 10% contamination” contains genome included in “>90% complete with less than 5% contamination”), but Figures shows separated set of genomes. In addition, line 142, MHQ are now defined as “≤ 5%” contaminated. The same issue rise with the definition of MQ MAGs, which could contain MHQ and HQ. As the selection of reliable genome is the foundation of all the following work, it would be useful and more convincing to have clearer definitions of the different genome subsets.

Indeed, by definition MHQ MAGs do include HQ MAGs. We now clarified this in the text by giving a clear definition of each MAG quality group, slightly different than MIMAG definitions (<https://www.nature.com/articles/nbt.3893>), in particular for MHQ as it corresponds to an intermediate quality level we defined in this manuscript. The MAG quality groups are defined as follows: HQ: ≥90% completeness and ≤5% contamination; MHQ: ≥75% completeness and ≤10% contamination; MQ: ≥50% completeness and ≤25% contamination. And we also corrected the Figure 1 legend to avoid any ambiguity.

b) Line 116: “we limited our analysis to HQ and MHQ genomes, which were of equivalently high-quality”: They are not of equivalently high-quality since the used quality thresholds are not the same. However, the scaling law residuals distributions are not significantly different for HQ and MQ (but is different of MHQ...).

That's correct, HQ and MHQ genomes are not of equivalent “high-quality”, rather their scaling law residuals distributions are not significantly different. We know rephrased this

sentence as follows: “*To ensure a sound and fair comparison between WGS and environmental genomes, we limited our analysis to MHQ genomes, which displayed a similar gene density as compared to WGS (Extended Data Fig. 1b). We showed that MHQ genomes did actually fit the same law as WGS genomes (Fig. 1b).*”

c) Mainly for MAGs, plasmids and other genetically mobile elements are generally not recovered during binning. What are the consequences for your analyses? Can you exclude this as a confounding factor when analyzing the genome sizes and CDS numbers (line 118-120: “This analysis also [...] with WGS genomes.”)? Is it possible to compare MAGs to WGS depleted from their plasmid-encoding contigs and other mobile elements?

This is a relevant comment highlighting a common problem in microbial genomics. Indeed, due to their variable copy number and sequence composition, plasmids and mobile genetic elements (MGE) are generally not well reconstructed nor captured for WGS genomes reconstructed from short reads (i.e., Illumina), and this is also true for MAGs (<https://www.ncbi.nlm.nih.gov/pmc/articles/PMC7660262/>). Thus, the bias due to the low recovery of plasmids and MGEs applies to all genome types considered and analysed herein. In addition, given the large differences observed in terms of genome size and number of CDS between MAGs and WGS, this is unlikely to be driven solely by missing plasmids and MGEs in all genome types. Nevertheless, the additional analysis suggested is interesting but would require significant work and analyses that we believe are out of the scope of this manuscript.

d) From Line 275 on, which quality of genomes is used? MHQ?

Yes, and this is now stated more clearly in the following sentence (line 118): “*We showed that HQ and MHQ genomes did actually fit the same law as WGS genomes (Fig. 1b), and thus limited all subsequent analyses to MHQ genomes only.*”

e) Line 487-490: during dereplication, which genome replicates is conserved? Usually it is the most complete and less contaminated, but this could have been adjusted by the user.

Thanks for this remark, during dereplication using dRep (with default parameters) we indeed kept the most complete and less contaminated genome/MAG as computed by checkM. We now further specified this selection in the methods section, as follows: “*This process allowed to identify and select the most complete and less contaminated genomes within each species cluster, which yielded 7,658 non-redundant species-level genomes...*”.

f) Genome networks were computed on which genome quality subset?

Genome-resolved networks were inferred using MQ genomes for which we could infer an abundance or activity measure, this is now more clearly stated in the methods section: “*Starting from all MQ genomes (N=7658), we defined the abundance of a genome in a sample by its overall metagenomic vertical coverage...*”.

2- At some instances, adding a short sentence or two explaining the strategies, the methods or the content of already published work would ease the reading, so that the reader does not need to scroll continuously to the methods section or find the reference to get the information necessary to understand or get convinced by the analyses.

Thank you for this comment. We now included in the main text some brief descriptions of the methods used as well as the content of already published work, when appropriate.

a) A broad readership will maybe not know how to read the residuals plots. A short explanation is probably necessary.

This is a good point. We now further describe the residual plots and how to interpret them in the main text. It reads as follows: “A linear least-squares regression was performed on a log-log scale, and the distribution of residuals for each category (difference between actual y-axis value and expected y-axis value) were compared. When significantly different, these distributions indicated that the two categories might not follow the same scaling laws.”.

b) Can you provide some information of the sequencing, albeit being already published? I was notably wondering about the sequencing depth in relation to the saturation, mainly for metatranscriptomics. Is it equivalent for all samples? If not, how does it impact the active sub-community analysis? As the more abundant genomes are also the more active ones, can you provide a short explanation on if the RNA was treated not to be DNA-contaminated to rule out this confounding factor?

Thank you for this important comment. The same sequencing depth was targeted for all samples for both metagenomics and metatranscriptomics. In addition, rigorous quality controls were applied for both DNA and RNA extracted from all samples. Additional information about sequencing is described in detail in a dedicated publication: Alberti *et al.* 2017 (<https://www.nature.com/articles/sdata201793>). Total RNA aliquots were treated with Turbo DNA-free kit (Thermo Fisher Scientific) and purified RNA was quantified with Qubit RNA HS Assay. The efficiency of DNase treatment was assessed by PCR, showing that the treatment was efficient in all checked samples. Thus, the observation that more abundant genomes are also usually more active (although there are exceptions to this general trend, see Fig.2b) is likely not attributable to DNA contaminations in RNA samples.

c) Line 196: Albeit pointing to the method here, giving the ‘activity threshold’ would make the concept/strategy clearer.

No activity threshold was defined nor applied per se. First, the occurrence of a genome/MAG was determined by its horizontal metagenomic coverage of minimum 30%. Second, its abundance was computed using its vertical metagenomic coverage normalised by its genome length. Finally, its activity corresponded to the ratio of its vertical metatranscriptomic coverage over its vertical metagenomic coverage. We now further detailed these computations in the main text (section ‘*Abiotic factors shaping genome community composition and activity*’), as well as in the methods within section ‘*Meta-omics profiling and associated environmental contextual data*’.

d) Line 251, SMETANA is introduced. I still did not get how were computed MRO and MIP scores from the SMETANA score, or even if it's the contrary.

Thanks for this question. The tool SMETANA enables the computation of multiple metabolic interaction scores that we used along this study. This is also the name the developers chose for one of the computed scores, hence the confusion. The MRO and MIP scores are community-wide and actually not directly related to the SMETANA sum score. Here, SMETANA was used to compute three distinct scores:

1. MRO (community-wide): the Metabolic Resource Overlap score quantifies how much species in a given community compete for the same metabolites.
2. MIP (community-wide): the Metabolic Interaction Potential calculates how many metabolites a given species can share with other members of the community to decrease their dependency on external resources.

3. SMETANA score (pairwise between two species): this score evaluates the probability of a specific cross-feeding interaction (2 species, one direction, one metabolite) by integrating three distinct metrics:
- SCS (species coupling score): measures the dependency of one species in the presence of the others to survive
 - MUS (metabolite uptake score): measures how frequently a species needs to uptake a metabolite to survive
 - MPS (metabolite production score): measures the ability of a species to produce a metabolite

SMETANA sum score (sum of the SMETANA scores in a given community) is employed by the original author and in our study as a community-wide version of the pairwise SMETANA score. We clarified this in the main text and we also now further describe these three scores in the methods section.

e) CarveMe was run without the time and resource consuming gap filling step. What are the consequence for the cross feeding model and analyses? Can this be discussed?

CarveMe was indeed run without the gap-filling step. The main reason for this was to avoid predicting potential false positive cross-feeding metabolic interactions. Given the incomplete nature of the genomes we used, this is a conservative approach, as without gap-filling we also likely miss potential true cross-feeding metabolic interactions. We now added a sentence to discuss this in the methods section.

3-Displays and their legend: albeit Figures are very eye-catching, some details could be modified to make them also really self-explanatory. Please find a list of instances where small modifications could be helpful to the readers.

a) Caption of the main figures and extended figures are sometimes incomplete: Fig 2B: what is "hor.cov"; The red dotted line is generally not described (cryptic in Fig 2b, and absent in Fig 3, extended Fig 1, 2 and 5). It is described in Extended Fig 7, but to depict something different;

In figure 3, the color code is confusing, with a color gradient that is not associate to the 3 colors used in the violin plot and in the legend. Where does stop the red and starts the grey? Please correct "N=?";

In figure 4, what does represent the insert? (residuals, as for the previous figures?). In panel b, how were the thresholds select? Please be consistent with the utilization of the abbreviation "PD" rather that "phyl.distance";

In figure 5: please give the stress of the NMDS in panel a. Please, expand the legend: How were the ellipses placed? What is the symbol size range from 2 to 12 indicating? What is the hierarchical clustering based on? What is the meaning of stars? Please homogenous the metabolite naming, notably for LalaDgluMdapDala, LalaDgluMdap, g3pe, Cys Glu, anhgm, etc... This comment also applies to the related Extended Figure 10.

Thanks for these remarks and useful suggestions that we now all carefully considered and addressed in improved versions of the figures and legends.

b) Extended Tables have no title, and some columns headers are not easy to understand. There is two Extended table 1 and two Extended table 2, which is pretty confusing.

We now added a title (see Supplementary information section) to each table and updated column names to make them more self-explanatory. We also removed redundant tables that corresponded to analyses not described nor discussed in the main text.

In Extend data Fig.1, panel d displays a reduced number of genomes compared to other panels. Can you explain?

The Extended data Fig.1 panel d displays a reduced number of genomes as it is limited to (co-)active genomes in photic samples.

In Extend data Fig 5, please explain $d > 2$ or $d < 2$;

In Extend Fig 5 and 6, explain in the legend what are the photic samples (in grey);

In extend figure 7 and 8: "size" of what?;

Please provide the y-axis unit in Extended data figure 9 and 11.

Thanks for these useful suggestions that we now all carefully considered and addressed in updated versions of the Extended Figures.

4-Networks, notably their construction, are not enough explained. Which type of correlations were computed? Were edges weighted (the correlation strength)?

Co-abundance and co-activity networks were reconstructed using FlashWeave (FW), which first infers a global correlation network resulting from significant partial correlation tests with Fisher's z-transformation ("sensitive" mode) between genome pairs co-abundance (or co-activity) profiles. Next, FW uses a local-to-global learning framework that implements conditional independence tests to detect and remove indirect associations within this network. Finally, the resulting network integrates only direct associations, whose edges are weighted by the partial correlation strength. We now added this more detailed description in both the main text and methods section.

How can you show us that "This genome-resolved co-activity network was significantly different than the corresponding genome-resolved co-abundance network" (line204-5).

A dedicated Extended Data Figure (Extended Data Fig. 4) illustrates the very small number of shared associations (3%) between the co-abundance and co-activity networks (panel a), although both networks display very similar distributions of edge weights, albeit overall stronger weights are observed in the co-activity network (panel b).

Can you discuss the presence or absence of (disconnected) modules in these networks?

Both co-abundance and co-activity networks reconstructed using FW were relatively small and sparse, with the co-abundance network counting 764 nodes and 1040 edges, and the co-activity network counting 787 nodes and 1205 edges. Both graphs were composed of a large connected component with a few small disconnected components. The co-activity graph was composed of one large principal component of 783 nodes and 2 small components of 2 nodes each. While the co-abundance graph was composed of one large principal component of 716 nodes and 19 small components counting from 2 to 4 nodes.

How variable is the activity network if the threshold of activity is moved away from 30%?

As specified above in response to comment 2c), we did not define an activity threshold but an occurrence threshold corresponding to a minimum of 30% horizontal metagenomic coverage (percent of the genome that is at least covered one time). Nevertheless, this threshold to define genome occurrences can eventually influence the network structure and topology. To investigate this, we compared three co-activity networks for different minimum

horizontal metagenomic coverage, i.e. for 25%, 30% (actual threshold used), and 35%. The edge intersections of these graphs is depicted in the Venn diagram below. Overall, a majority of edges are shared between all three graphs, although (as anticipated) we can observe an increase of unique edges for the cov.25% graph, that is when decreasing this threshold. This is expected as lowering this threshold will increase the number of occurring genomes and thus the number of new edges linking them.

5-General questions on the presented results:

a) The discussion from Line 120 to Line 126 is not clear to me. How did you select the 4 KEGG categories out of the Extended Table 2? A lot more categories seems to meet the statistical objectives. Why focusing on KEGG rather than COG? What about displaying a dotplot for every category so the reader can check out?

We focused on the functional potential comparison between genome types using the KEGG metabolic categories as the primary focus of this study is to predict potential community metabolic interactions. It is correct that most (75%) of the metabolic categories considered display a higher or lower prevalence in WGS as compared to MAGs (Supplementary Table 3). Thus, we decided to select and discuss specific metabolic categories that are playing a role in biotic interactions. We now made this clearer in the discussion of these results in the main text as follows: *“we showed that this adaptation has differentially impacted a majority of metabolic functions (75%) within uncultivated genomes (MAGs and SAGs), with notable increase potential in metabolic functions likely playing a key role in mediating biotic interactions, such as...”*.

b) Line 216-219, you discuss the potential niche overlap (and thus possible competition) between short PD taxon. As you have the transcriptomic data for them, can you verify your assertion checking which genes are expressed and if they cover the same function? It would be much more convincing.

Thank you for this useful suggestion. To complement our analyses, we computed a functional (Jaccard) distance based on KO genes expression profiles between genomes. This distance is equivalent to the KO-based functional distance of Fig.3a but includes expression information by calling the expression of a KO gene if expressed in at least 10 samples. Using this “expressed KO distance”, we reproduced Fig.3a and observed that co-

active genomes also tended to be functionally closer in expression than expected at random, although to a lesser extent than measured by the “encoded KO distance”, that is in terms of functional potential. We now included and discussed this analysis in Fig.3.

c) In co-active communities and from metabolic reconstruction, what is the proportion of ‘selfish’ taxa and of ‘altruistic’ or ‘generous’ taxa? Is there some trends?

Thank you for this interesting question. To address it we classified genomes in co-active communities, involved in predicted metabolic exchanges with a minimum SMETANA score of 0.5, into three types: i) Mostly Donor, ii) Mostly receiver, or iii) None. The categories 'Mostly Donors' and 'Mostly Receivers' include genomes that either provide or receive all exchanged metabolites, or alternatively, provide or receive at least n-1 metabolites of the n exchanged metabolites, where n>2, while “None” encompassed the remaining genomes. This analysis (see figure below) revealed a relatively even proportion of mostly donor (31%) and mostly receiver (36%) genomes. When investigating the taxonomy of these genomes, we can report two interesting observations: i) genomes of the order Pelagobacterales appear to be mostly receivers, which is coherent with their usually oligotrophic (scavenger) lifestyle and their unusually small genome size; ii) genomes of the order Flavobacteriales appear to be mostly donors, which seems to be coherent as well as species from this group are often associated with marine snow and marine phytoplankton blooms, and usually correspond to copiotrophs common in environments with greater nutritional opportunities.

d) Are the HCP community described around line 326-330 sharing some environmental specificities? Are they derived from region with a high or a low productivity? Are some physico-chemical parameters specific to the related sampling area?

Thank you for these interesting questions. In response, we investigated the biogeography of HCP communities (see map below). This analysis illustrates that HCP communities are detected at global scale (i.e., across all samples) but that they all display different degrees of regionalization (i.e., no HCP community is detected across all samples). When investigating the link between HCP communities and associated physico-chemical parameters (e.g., temperature, latitude, Chlorophyll A, NPP – see figures below), we observed a specificity to temperature related to the latitudinal organisation of these communities, as previously reported using *Tara Oceans* amplicon sequencing data (Chaffron et al. 2021). HCP communities are also reported for relatively different concentrations of Chlorophyll A and different levels of Net Primary Production (VGPM model). Nevertheless, it should be noted that most ocean regions sampled during *Tara Oceans* corresponded to low productivity /

oligotrophic regions. We now integrated this novel analysis as an additional Extended Data Figure.

b

e) Genomes and active genomes observed in less than 10 samples were removed. Have you checked their depth of coverage? Are you removing only low abundance / low activity taxon? Or are you also removing taxon with local high abundance/activity?

Thank you for these questions. We can expect that filtering out (active) genomes observed in less than 10 samples will remove locally high abundant and locally high active genomes. To address these questions we compared the genome-wide abundances and activities of low (<10 samples) and high (≥ 10 samples) prevalence genomes, in samples ($n=104$) for which we have both metagenomics and metatranscriptomics data (see figure below). In abundance (metagenomic coverage divided by genome length then multiplied by 10^6 , TSS

normalized), high prevalence genomes are significantly more abundant compared to low prevalence genomes. While in activity (metatranscriptomics coverage divided by genome length then multiplied by 10^6 , TSS normalized, divided by the abundance above), high prevalence genomes overall displayed (significantly) slightly smaller activity levels as compared to low prevalence genomes.

Minor comments

1 - Line 102 “HQ and MHQ MAGs were not significantly different from WGS genomes in terms of gene density (Supp. Table 1)”: I did not find the statistic for this assertion in the mentioned display.

Thanks for this comment. The gene density or median number of CDS per 1M bp (median_cds_per_1M_bp) information in the Supp. Table 1 summary is provided by genome resource but not by genome type (WGS, MAG, SAG). We now provide this gene density information by genome type in Supp. Table 2 (previously Supp. Table 1).

2 - Line 107: what is a “high-level functional category”?

A high-level functional category refers here to broad functional categories regrouping related gene functions, such as the Clusters of Orthologous Genes (COGs) 17 functional categories (e.g., replication, recombination and repair; nucleotide transport and metabolism) or the KEGG BRITE Functional Hierarchies (e.g., Energy metabolism; Metabolism of cofactors and vitamins). We now specify what we refer to in the main text and also describe what high-level functional categories we used in the methods section.

3 – Line 118: please correct the typo “...that HQ/MHQ MAGs and were systematically...”
Done, thank you.

4 - Line 124 “uncultivated genomes”: please rephrase, as no genomes are cultivable yet ;-)
Thank you, we now rephrased this and refer to environmental genomes.

5 - Line 145 chlorophyl versus chlorophyll

Now corrected, thanks.

6 – In Figure 2B, the titles “abundance” and “activity” looks like axes titles. Can they be slightly moved?

We now moved both titles to the upper right corner of each panel.

7 – Line 210 “The co-activity network revealed a larger number of significant positive associations across large phylogenetic distances (PD), while negative associations were mainly observed between phylogenetically distant genomes ...”: Isn't it contradictory? From the figure, I understood that positive associations are rather between population of short phylogenetic distances.

In Fig.3a, red points correspond to negative co-activity associations while blue points refer to positive co-active associations. On the right panel upper boxplot, it becomes more visible that significant positive associations cover a wide (likely bimodal) distribution of phylogenetic distances, while significant negative associations are restricted to a narrower distribution of larger phylogenetic distances. In addition, these distributions of phylogenetic distances for negative and positive associations are significantly different (Mann-Whitney U-test with Bonferroni correction, $p=1.494 \times 10^{-30}$).

8 – Line 222 cites Fig.1, but this figure does not show the abundance of genome versus their size.

That's correct, Fig.1a only shows that the majority of genomes detected in photic samples (outer circle) correspond to MAGs, and Fig.1b shows that MAGs are overall smaller in size. We now modified the sentence as follows: “... *and despite the fact that most abundant and active genomes detected in photic samples corresponded to MAGs (Fig. 1a) overall smaller in size (Fig. 1b).*”.

9 – Please tone down Line 253-4: “... to predict metabolic interaction potential and reveal [possible] metabolic exchanges and cross-feedings...”

Done, thank you.

10- Line 257: what does mean “community species”

This is to refer to the ensemble of species within a given community, we reformulated the sentence as follows: “... *and the Metabolic Interaction Potential (MIP) score quantifies how many metabolites can be shared between species to decrease their dependency on external resources*”.

11- Line 267 “...which we thus normalised by community size ...”: is it meant co-active communities or the total community?

We meant normalised by the size of the co-active community under consideration, we now specify it in the sentence.

12 – Last sentence of the paragraph 260-274, and the first sentence of the following section (line 275-277) reads contradictory. Or should “large” line 275 rather reads divers/various?

Thanks, indeed “large” (l.275) should rather reads “diverse” here, we corrected.

13- Line 275-6 “...among co-active genomes (Fig. 3a) and communities...”: which community are you referring to?

Here we refer to co-active genomes and resulting communities of co-active genomes, we now better specify this.

14 – Line 278-9 “largely composed of genome communities”: please rephrase, what are ‘genome communities’?

Here we refer to communities of co-active genomes, we rephrased that sentence to make it clearer.

15 – Line 317-8 “Both HPD-HCP and LPD-HCP communities were predicted to have a higher potential exchange in specific metabolites as revealed by a NMDS ...”: isn’t it the definition of HCP (high cross feeding potential)?

HCP communities are indeed defined as communities displaying an overall higher predicted potential for metabolic interaction (Fig.4b), but the NMDS analysis referred to here (Fig.5ab) allows to detail and identify which underlining specific metabolites are predicted to be exchanged within these HCP communities (Fig.5c).

16 - line 319 “large metabolic categories”: do you mean broad?

Thank you, we meant broad and adjusted the sentence.

17 – Line 382 “we investigated their graph centrality...”: who is ‘their’. It reads like if it was metabolites.

This sentence is indeed misleading, we investigated the graph centrality genomes (not metabolites), we now rephrased to make this clear.

18 – Line 386 and line 387 “genome donors” and “non-donor genome”: please rephrase, it is not genome that encode the metabolic potential of sharing, and not the genome that is providing the metabolite

We now replaced genome by species along the text in this specific section.

19-Line 492 “GTDB-TK”: please check for the caps.

This is now corrected to GTDB-Tk.

20 – Line 572 “... and activity (N=902 genomes observed in at least 10/71 samples).” should read “... and activity (N=902 *active* genomes observed in at least 10/71 samples).”

Now corrected, thank you!

21 – Line 929-930 “We used samples with both metagenomics and metatranscriptomics available to compute genome-wide co-activity.”: can this information be in the main text?

We now also added this important information in the main text (first paragraph of section ‘Biotic drivers of genome activity community structure’).

22 – Line 949-950 “Genomes in the co-activity network are significantly smaller both in size and number of CDS”: I don’t see that.

This is particularly visible from the density plots on both axes for genome size (x) and number of CDS (y), in both panels a and b distributions for genomes in the co-active network are significantly smaller both in size and number of CDS (Mann–Whitney U test, p -value = 1.84×10^{-45} and 3.22×10^{-46} , respectively).

Reviewer #2 (Remarks to the Author):

This is an exciting study that takes a significant step beyond the ubiquitous abundance correlation networks so common in microbial ecology literature. By leveraging information from extensive genome database and environmental omics data sets, the authors provide a roadmap for functional and metabolic analysis. They aim to expand upon co-occurrence networks by utilizing genome-resolved metagenomics data sets together with meta-transcriptomic data to uncover activity in transcription. The authors rely on genome scaling laws to characterize the functional content of coactive genomes, identifying functional gene categories that might drive metabolic dependencies in the community. By employing genome metabolic models, they uncover potential cross-feeding interactions

This study addresses an important problem in marine microbial ecology, specifically how microbial communities are assembled depending on their functional capabilities. The authors are particularly interested in developing a mechanistic understanding of how metabolic auxotrophy constrains community composition and assembly. This is an exciting study that does much to go beyond the abundance correlation networks that are so ubiquitous in microbial ecology literature now. It does this by starting to leverage the genomic information in our genome database and environmental omics data sets. The field will likely find this manuscript useful in providing a road map to do such functional and metabolic analysis.

We wish to thank reviewer 2 for his useful and constructive comments, and appreciate the feedback about the relevance of our ecosystems biology approach to go beyond species correlation networks by combining genome-resolved meta-omics with community metabolic modelling to predict potential cross-feeding interactions in bacterioplankton communities. Below we detail specific actions we have taken to address his comments. All modifications are **highlighted in yellow** in the revised manuscript.

Major comments

I completely agree that functional, metabolic and ecosystem modeling approaches, like the one taken here, are needed to understand the complex interactions and emergent properties of these microbial communities from the genome content and how they fit into ecological and biogeochemical roles. However, my main concerns with the manuscript are its sheer density and the fact that the authors have packed a lot of information into a relatively small space, making the logic sometimes difficult to follow. This issue seems to stem from having to bounce between results and methods or potentially descriptions in the figures. The manuscript would likely be more readable if a figure guided the reader through the progression of the different analyses and metrics used.

Thank you for this comment, we now did our best to improve this point by further describing our methodology along in the main text. In addition, we created a novel Extended Data Fig.1 describing schematically the overall approach, workflow, and methodology. We hope this addition will ease the understanding of our ecosystems biology approach.

My other major concern was the choice to use an external genome data set to map the TARA omics data to, as the percentage of TARA reads mapped to this database was relatively low. This raises a methodological concern about the choice to use the genome

catalog versus trying to use assembled genomes directly from the Tara oceans data set. It's not clear why this decision was made. I would expect if you would have used the assembled genomes from Tara you would have had a higher MG and Mt read mapping. The authors should provide an explanation for why they chose to go with the external genomes versus the Tara assembled genomes and clarify this in the text.

Metagenomics and metatranscriptomics reads mapping are indeed relatively low. This is a common problem encountered in genome-resolved metagenomics, mainly due to the difficulty to reconstruct high strain diversity and/or low-abundant genomes. Here, we actually did integrate both WGS genomes and MAGs from other studies as well as *Tara* Oceans data assembled MAGs. This is specified in the methods section: "*This collection of well-documented genomes was complemented by 5,319 MAGs assembled from four distinct studies, namely: Parks et al. 2017 (N=1,765; downloaded from EBI), Tully et al. 2017/2018 (N=2,597; downloaded from EBI), and Delmont et al. 2018 (N=957; downloaded from FIGSHARE).*". To address this point we now also added this information in the main text.

Overall, my suggestion is for revisions to increase the clarity and digestibility of the manuscript, particularly given the density of analysis conducted. This will ensure that this important work can be more accessible to the many who will be interested in it.

Thank you for this important comment. We hope that the inclusion of the novel Extended Data Fig.1 and its description, as well as further descriptions of the methodology along the main text will help in successfully addressing this comment.

Line 508: "We used scaling laws as a framework to characterise the functional content of our genomic database"... This is such a key piece of the analysis conducted, but it's not very clear what it actually means. The genomes have their functional content based on their genes and resulting KEGG and eggnoG annotations. What and how were the scaling laws used to do. Apologies if this is something that should be obvious, but I was having a hard time making the specific connections from the way its described in the manuscript.

We now provide further description and explanation about the genomic scaling laws framework within the methods section. In addition, we also now briefly describe the concept in the main text.

Detailed Comments

Line 118: typo, "and" at the end of the line.

Now corrected.

Line 119. Is this observation supposed to be able to be seen in figure 1B? If so it seems the MAG span the genome sizes in the figure.

Yes this observation is particularly visible from the density plots on both axes for genome size (x) and number of CDS (y) in Fig.1b. Both x and y distributions for MAGs (in orange) are clearly shifted towards smaller genomes and lower number of CDS as compared to WGS genomes (in black).

Line 122 - 126: There is a lot here. It seems like an important part of the analysis but it's only briefly described how this was done, with most of it in the supplementary information. Would it be helpful to expand on this a bit to help clarify to the reader what's going on. Also the

direction of the trend is not specifically mentioned. For example, I assume that decrease metabolic potential for xenobiotics and polyketone aids is in smaller genomes, but this is not explicitly stated.

As stated above, we now provide further details in both the main text and the methods section about why and how we used scaling laws to compare encoded metabolic potential within genomes while taking into account their size.

Line 155: There is a lot in the results that the authors force the reader to bounce back and forth between the methods and results. For example the genome-wide activities. It would be much easier to read and digest the study if they could include a simple description of what the genome-wide activity metric is when it is first mentioned. Or how what the co-activity to co-abundance comparison metric is.

In response to a similar comment by reviewer 1, we now further described in the main text what are genome-wide activities and also how they were computed.

Line 157. What is the metric that you are using for the PCA in terms of species abundance? Coverage of mapped reads to the genomes? I appreciate that there are lots of analyses in this study and the importance of being concise, but often this makes it difficult to follow exactly what is happening.

Thanks for this comment and question. The metrics used in both Principal Coordinates Analyses (PCoA) were the genome-wide abundance and activities, which we now further described in the main text as stated above.

Fig 2. Does the differences in sample coverage between the MG and MT datasets lead to biases? It's a bit hard to tell, but it looks like there is less of North Atlantic and Mediterranean samples in the MT versus the MG. This could cause biases in the environmental factors identified as correlating with the PCA axis.

There is indeed a different sample coverage for metagenomics and metatranscriptomics data, with 107 samples across 64 stations for metagenomics data, 118 samples across 81 stations for metatranscriptomics data, and 71 samples across 45 stations for which we had both (see methods). Nonetheless, our primary goal here was to maximise the amount of data integrated to best reveal abiotic factors shaping genome community abundance and activity. For the activity-based PCoA we integrated all samples for which we had both data available, which thus correspond to a subgroup of samples from the abundance-based PCoA. This is now clearly specified in the Fig.2 legend.

Line 195: IS this activity different than the one mentioned above?

This genome-wide activity is the same as the one mentioned above, and we now specify it in the main text.

Line 206: This could use clarification. Do you mean a small fraction shared between the co-activity and co-abundance networks?

There is indeed a very small fraction of shared edges (3%) between both co-activity and co-abundance networks. This is visualized using a Venn diagram in Ext. Data Fig.4 panel a.

Line 288: The gini coefficient is first mentioned here but not explained what it means until the methods.

Thanks for this comment, we now introduce and explain this index in the main text as well.

Line 330 typo “quote zooming in on”

Now corrected, thanks.

Line 339 it's not clear the distinction between these two community comparisons what the difference is between “enriched” and “predicted”.

This is indeed misleading. Here, “predicted” refers to the metabolic exchanges predicted, while “enriched” refers to the same metabolic exchanges predicted several times within HCP communities. We now rephrased this sentence and referred to “enriched” metabolic exchanges as “significantly more prevalent” exchanges.

The example with *Prochlorococcus* and other taxa was extremely useful. Could this be expanded for other organisms in your analysis. This section likely provides the most concrete take away takeaways from the entire manuscript.

All predicted metabolic exchanges are available in Supplementary Table 6. We could not automate (yet) the visualisation of these predicted exchanges but we are planning to build a resource including maps of predicted community metabolic exchanges. Nevertheless, we represented predicted metabolic exchanges for three small HCP communities to highlight specific exchanges of sugars, AAs, and B vitamins (see figure below). Graphs represent predicted metabolic exchanges (SMETANA score ≥ 0.5) between genomes of communities ‘coact-MHQ-041’, ‘coact-MHQ-056’, and ‘coact-MHQ-096’. Exchanges of several amino acids, B1 vitamin (thiamin), and monosaccharides were predicted between these genomes.

coact-MHQ-041-mcl1.5 coact-MHQ-056-mcl1.5

coact-MHQ-096-mcl1.5

Figures sometimes need more detailed explanations especially since they are so dense. For example figure 5a: what are the circle sizes supposed to represent? Or in figure 5B what do the asterisks mean?

Thank you for this comment, we now further detailed the description of Fig.5 in the legend.

Reviewer #3 (Remarks to the Author):

Identifying how active microbes interact with each other in a community is essential to predict microbial community composition, which has a crucial impact on the element cycling and primary production of the oceans. Using Tara Oceans meta-omics data, Giordano et al predicted inter-lineage associations between co-active microbial communities of the euphotic ocean, which was interpreted as conserved cross-feedings of cellular metabolites, particularly costly amino acids and group B vitamins. In addition, the authors also found that microbes with high cross-feeding potentials were overall smaller in genome size, thus proposing that genome streamlining and metabolic auxotrophies were central joint mechanisms shaping the assembly of bacterioplankton communities in the epipelagic ocean.

Overall, this is a high-quality manuscript with sufficient novelty and broad community interest.

The content is well organized with an adequate introduction, detailed method description, accurate result interpretation, insightful discussion, and appropriate references. However, as currently submitted, there are two major concerns that should be carefully addressed.

We wish to thank reviewer 3 for his useful and constructive comments, and appreciate the positive feedback about the relevance and quality of the work.

Below we detail specific actions we have taken to address his comments. All modifications are **highlighted in yellow** in the revised manuscript.

Firstly, the authors acknowledged that the prediction of metabolic exchange within co-active communities should be further validated in the lab through co-culture experiments, which I believe is well beyond the scope of the current manuscript. However, the current prediction is not convincing without further analysis of the exchange mechanisms between donors and receivers/non-donors. I kindly request the authors investigate the transporters responsible for specific amino acids and group B vitamins in co-active communities. It would be beneficial to confirm if these transporters are present/enriched in the non-donor genomes and actively transcribed.

Thank you for this useful comment and pertinent suggestion. We now investigated the transporters responsible for amino acids and B vitamins predicted exchanges within co-active communities as well as their occurrence and activity in co-active genomes (see figure below). Transporter-associated reactions were directly extracted from CarveMe reconstructed models and classified into 9 different types of transporter mechanisms and two directions (import or export). Reversible transport reactions were duplicated and counted in both directions. This analysis, now integrated as a new figure (see below and Extended Data Fig. 13) revealed a vast majority of ABC systems for import reactions of specific AAs and B vitamins predicted exchanged, while more diverse transporter types (e.g., diffusion, proton antiport) are responsible for export reactions (panel a), across the four distinct community types (panel b). Activities (abundance-normalized expressions) were obtained by exploiting the so-called gene-reaction-rule encoded within the CarveMe reconstructed models. These rules often involved the expression of multiple genes to have a functional transporter protein, but since metagenomics and metatranscriptomics data are known to be sparse (especially at the gene level), we considered a transporter expressed if at least one of its components was actively transcribed. Overall, we were able to identify AAs and B vitamins transporters and associated reactions, linked them with actual CDS present in our

co-active community genomes, and detect a transcriptional activity of these in multiple samples of our dataset.

It is possible that some of the observed cross-feeding is due to substrate-based metabolic partitions where each community member independently utilizes distinct substrates released or degraded from the same source.

This is correct and we now discussed in the main text the fact that some metabolic cross-feeding predicted herein may be due to substrate-based metabolic partitioning.

Secondly, the claim that genome streamlining and metabolic auxotrophies were the central mechanisms shaping the assembly of microbial communities in the surface ocean may be overstated. Black Queen (BQ) processes may lead to mutualistic interactions and genome reduction of non-donors if they have sufficiently large effective population sizes (N_e), otherwise, BQ gene loss and genetic drift may ultimately lead to obligate symbiosis (Giovannoni et al., 2014 ISME J). Since all the non-donors discussed in this study are free-living, they either have a large N_e or are only losing costly traits instead of undergoing

significant streamlining evolution. In addition, genome streamlining is more prevalent in nutrient-limited environments, suggesting abiotic factors shouldn't be ignored in the discussion.

Thank you for this important comment. We agree that our overall interpretation “that genome streamlining and metabolic auxotrophies are central mechanisms shaping the assembly of microbial communities in the surface ocean” is overstated due to a wrong formulation. We now tone down our interpretation by reformulating the text in both abstract and main text. Here, we nevertheless would like to suggest that these mechanisms may act jointly in (oligotrophic) surface ocean bacterioplankton communities. The points raised here by reviewer 3 are actually very interesting and rather tend to further support our interpretation, as surface ocean bacterioplankton will indeed tend to have a large N_e , and because most *Tara* Oceans samples have been collected in nutrient-limited (oligotrophic) ocean regions. We now further discuss these points in the discussion. Notably, it is important to note that our results are indeed limited to free-living bacterioplankton communities in tropical and temperate ocean regions. Nevertheless, abiotic factors, such as temperature, are also conjointly impacting genome size in the ocean microbiome.

Minor comments:

1. L118 and L142, has the genome size been normalized by its completeness?

Thank you for this question. Here, neither the genome size nor the number of CDS were normalised by completeness. Although, this would not impact the genomic scaling law analysis (l.118), it could influence the genome size comparison (l.142). Thus, to address this question, we repeated the Fig.1b analyses after normalising genome size and the number of CDS by the completeness. This resulted in a small decrease of significance when comparing genome size, but MAGs still displayed overall smaller genomes as compared to WGS. Also, as anticipated, it had no impact on the scaling law analysis as both axes were normalised by genome completeness.

2. L156, have rRNA reads been depleted or not?

Yes, prior to metatranscriptomics sequencing, cDNA synthesis of total RNAs was performed by a random priming approach preceded by a prokaryotic rRNA depletion step. This is described in Alberti et al. 2017 (<https://www.nature.com/articles/sdata201793>), which we cite in the methods section, where we now directly added this information.

3. Fig 2c, it would be better to have the regression lines in the subplots.

We decided not to add the regression lines to each subplot due to the sometimes low correlation coefficients observed, and also to avoid a potential over interpretation on the linearity between environmental parameters and principal components.

4. L215, how do you define “functionally related”?

As described in Fig.3 legend, we defined a functional distance (or relatedness) between genomes by computing KO-based functional Jaccard distances between genome vectors of KO gene presence/absence. We now added this information in the main text as well as in the methods section.

5. Fig 5a, what does the point size mean?

The point size in Fig.5a represents the size of each community in number of genomes. This information is now added to the figure legend.

6. Fig 5b, what's inside of "uncategorized"?

We attempted to classify each metabolite into metabolite categories (e.g., amino acids, carboxylates) using the MetaNetX and MetaCyc databases (see methods). In this process, a number of metabolites could not be assigned to any metabolite category and were thus dumped as "uncategorized".

7. "p" should be italicized in "p-value" at multiple locations.

Thank you, this is now corrected.

8. Similarly, the multiply symbol should be used instead of "x" at multiple locations.

Also corrected now.

Reviewer #1 (Remarks to the Author):

Thank you for this diligent revision.

New analyses are convincing and I do not have additional major comment. The only typo I picked up is line 1198, where x-axis should reads y-axis.

Reviewer #2 (Remarks to the Author):

The authors have sufficiently addressed my original concerns.

Reviewer #3 (Remarks to the Author):

I'm pretty satisfied with the authors's responses to my previous comments. This manuscript can be accepted for publication after fixing the following minor issues:

Lines 370-372: the authors argued that these predicted metabolic exchanges are ancient and evolutionarily conserved. Actually, some of the metabolites might be metabolic wastes (like aromatics), meaning they could be exported to the environment as a stress response, but not to exchange anything. This is even true for some amino acids when they're overproduced due to variations in nutrient availability or metabolic imbalances within the cell.

Line 390: should be "nutrients"

Line 403: should be "AAs"

Line 421: should be "genomic"

Line 476: "recent" and "recently" are duplicated

Line 515: "p" in "p-value" should be italicized

Reviewer #3 (Remarks on code availability):

Yes, the code can be found under <https://gitlab.univ-nantes.fr/ecosysmic/EcoSysMic-analysis> repository, and within each directory, usually a Jupyter notebook with codes and figures can be found.

REVIEWER COMMENTS

We would like to thank again all three reviewers for their useful and constructive comments. We now have carefully addressed their final comments in the revised manuscript.

Reviewer #1 (Remarks to the Author):

Thank you for this diligent revision.

New analyses are convincing and I do not have additional major comment. The only typo I picked up is line 1198, where x-axis should read y-axis.

Thank you for spotting this mistake that is now corrected in the revised manuscript.

Reviewer #2 (Remarks to the Author):

The authors have sufficiently addressed my original concerns.

Reviewer #3 (Remarks to the Author):

I'm pretty satisfied with the authors's responses to my previous comments. This manuscript can be accepted for publication after fixing the following minor issues:

Lines 370-372: the authors argued that these predicted metabolic exchanges are ancient and evolutionarily conserved. Actually, some of the metabolites might be metabolic wastes (like aromatics), meaning they could be exported to the environment as a stress response, but not to exchange anything. This is even true for some amino acids when they're overproduced due to variations in nutrient availability or metabolic imbalances within the cell. We agree with this comment and we now included an additional sentence to discuss this possibility, which reads as follows (l.304): "...*although we cannot exclude some of these metabolites might be metabolic wastes that could be exported to the environment as a stress response.*".

Line 390: should be "nutrients"

Line 403: should be "AAs"

Line 421: should be "genomic"

Line 476: "recent" and "recently" are duplicated

Line 515: "p" in "p-value" should be italicized

Thank you, all these mistakes are now corrected in the revised manuscript.

Reviewer #3 (Remarks on code availability):

Yes, the code can be found under <https://gitlab.univ-nantes.fr/ecosysmic/EcoSysMic-analysis> repository, and within each directory, usually a Jupyter notebook with codes and figures can be found.

Indeed, all the code for processing and analysing the data are available at <https://gitlab.univ-nantes.fr/ecosysmic> and <https://gitlab.univ-nantes.fr/ecosysmic/EcoSysMic-analysis> (a sub repository). We now added a readme to both repositories and also documented the code usage and all folders content.